# Interactions between fibroblastic reticular cells and B cells promote mesenteric lymph node lymphangiogenesis

Lalit Kumar Dubey [1], Praneeth Karempudi[1], Sanjiv A. Luther[2], Burkhard Ludewig[3] & Nicola L. Harris[1]

Lymphatic growth (lymphangiogenesis) within lymph nodes functions to promote dendritic cell entry and effector lymphocyte egress in response to infection or inflammation. Here we demonstrate a crucial role for lymphotoxin-beta receptor (LTβR) signaling to fibroblastic reticular cells (FRCs) by lymphotoxin-expressing B cells in driving mesenteric lymph node lymphangiogenesis following helminth infection. LTβR ligation on fibroblastic reticular cells leads to the production of B-cell-activating factor (BAFF), which synergized with interleukin-4 (IL-4) to promote the production of the lymphangiogenic factors, vascular endothelial growth factors (VEGF)-A and VEGF-C, by B cells. In addition, the BAFF-IL-4 synergy augments expression of lymphotoxin by antigen-activated B cells, promoting further B cell–fibroblastic reticular cell interactions. These results underlie the importance of lymphotoxin-dependent B cell–FRC cross talk in driving the expansion of lymphatic networks that function to promote and maintain immune responsiveness.

[1] Global Health Institute, School of Life Sciences, École Polytechnique Fédérale de Lausanne (EPFL), Lausanne CH-1015, Switzerland. [2] Department of Biochemistry, University of Lausanne, Lausanne CH-1066, Switzerland. [3] Institute of Immunobiology, Kantonsspital St. Gallen, St. Gallen CH-9007, Switzerland. Correspondence and requests for materials should be addressed to N.L.H. (email: nicola.harris@epfl.ch)

Lymphatic vessels play an important role in tissue fluid homeostasis and promote the drainage of fluids and cells from tissues to the lymph node (LN)[1, 2]. Although lymphatic vessels develop during embryonic life, lymphangiogenesis (defined as the formation of new vessels) can occur in adults under various conditions, including wound healing, cancer, and inflammation. Intranodal lymphangiogenesis is crucial for promoting dendritic cell (DC) entry to[3, 4], and lymphocyte egress from[5, 6], the draining LN. Emerging evidence suggests lymphatic endothelial cells (LECs) can also directly regulate immune responses[7] by promoting T-cell tolerance against self-antigens[8, 9] and maintaining anti-viral T-cell responses through the capture and archiving of viral antigens[10]. Thus, understanding how inflammation regulates intranodal lymphangiogenesis is essential for our understanding of adaptive immune responses.

Lymphangiogenesis occurs via a vascular endothelial growth factors (VEGF)-dependent process that involves sprouting, migration, proliferation, and tubule formation by LECs[11]. Lymphatic growth is well known to require VEGF-C interactions with VEGFR-3[2], and a role for VEGF-A in promoting inflammatory lymphangiogenesis has also been reported[3, 12]. Although the roles of VEGF-A and VEFG-C are well established[2, 12–14], the contribution of other cytokines, or of stromal vs. hematopoietic cells, in regulating intranodal lymphangiogenesis remains unclear[15]. Recent studies have demonstrated an important function of T cells in exerting an anti-lymphangiogenic role via IFN-γ secretion[16, 17], whereas a pro-lymphangiogenic role of B cells has been demonstrated, but is context dependent[3, 12, 13].

The mesenteric LN (mLN) maintains an active homeostasis during steady state conditions but quickly enlarges in response to infection with intestinal pathogens[18–21]. The factors governing mLN lymphangiogenesis have not been characterized. We addressed this question using the model murine helminth, *Heligmosomoides polygyrus* (*Hp*), which has an infectious lifecycle limited to the intestine[22]. *Hp* infection elicits a strong type 2 immune response in the draining mLN[21] and we have previously reported that protective immunity requires lymphotoxin-dependent stromal cell remodeling and the formation of new B-cell follicles[19].

In this study we have used *Hp* as a tool to compare the interactive behavior of stromal cells within organized lymphoid tissue in which adaptive immune response develop. Using immunofluorescence staining combined with deep tissue imaging we now show that *Hp* infection results in extensive mLN lymphangiogenesis that correlates with enhanced DCs entry. mLN lymphangiogenesis was driven by a complex interplay between inflammatory cytokines, fibroblastic reticular cells (FRCs) and B cells. Lymphotoxin-dependent activation of mLN FRCs promoted the production of B-cell-activating factor (BAFF), which synergized with the type 2-cytokine interleukin-4 (IL-4) to activate VEGF production by B cells and to drive the proliferation of LECs. Our findings provide a novel mechanistic view of mLN lymphangiogenesis and demonstrate a previously unidentified function for FRC-derived BAFF, which provides the necessary signal for LEC expansion by programming B cells within the secondary lymphoid organs.

## Results

### Intestinal helminth infection elicits extensive mLN lymphangiogenesis. *Hp* is a enteric murine nematode that exhibits pathogenic traits and serves as an excellent model for studying Th2-driven immune responses[23]. The helminth-infected host requires B cells and CD4+T cells for the development of sterilizing immunity and resistance[19, 24]. However, the impact of these macro intestinal pathogens on the draining lymphoid tissues has not been studied in detail. Moreover the migration of antigen-presenting cells from the intestine to the draining mLN via the lymphatic vasculature is necessary for eliciting effective intestinal immunity[25]. To determine whether intestinal helminth infection could promote mLN lymphangiogenesis we examined the entire chain of the draining mLN of naive and *Hp*-infected mice and then visualized the lymphatic vessels by staining sections with an antibody against the LEC-specific marker, lymphatic vessel endothelial hyaluronic acid receptor-1 (LYVE-1). Deep tissue imaging of stained sections revealed that lymphatic vessels in naive mLN formed a dense lymphatic network immediately below the subcapsular sinus region (SCS) and within the medullary sinuses, but not within the central paracortical regions (Fig. 1a and Supplementary movie 1). Infection with *Hp* drove the dramatic growth of new lymphatic vessels, which were apparent by 6 days post infection (dpi), but which became more pronounced by 12 and 21 dpi (Fig. 1a–c and Supplementary Fig. 1a–d and Supplementary movies 1 and 2). New lymphatic vessels were observed to extend deep into the paracortical zone of the mLN, as visualized in vibratome slices (ranging from 200 to 2000 μm) obtained from central part (Fig. 1b and Supplementary Fig. 1e and Supplementary movie 3). Increased lymphangiogenesis in *Hp*-infected mice was also confirmed by an increased expression of *Prox-1* mRNA transcripts (Fig. 1d), together with an increased expression of *Vegf-a* and *Vegf-c* mRNA transcripts (Supplementary Fig. 1f, g) and protein levels (Supplementary Fig. 1h, i) in the mLN. These data indicate that chronic intestinal helminth infection is associated with extensive mLN lymphangiogenesis.

### Lymphatic vessels form close contacts with B cells and FRCs. We next examined the location of LECs relative to the newly developed B-cell follicles that we previously reported formed following *Hp* infection[19]. The extensive network of lymphatic vessels observed next to the SCS of mLNs from naive mice were observed to surround B-cell follicles found in this region (Fig. 1e). Following *Hp* infection this network expanded to additionally surround the newly formed B-cell follicles located in the paracortical region (Fig. 1f and Supplementary movie 4 and 5). For both naive and infected mLNs, LECs formed a "cup"-shaped structure that encapsulated, and possibly supported, the B-cell follicle (Fig. 1e, f and Supplementary movie 5 and 6). As FRCs of infected mice typically form close interactions with B cells we also performed a co-staining of LECs and FRCs. FRCs and LECs share common surface markers such as Podoplanin (PDPN), whereas LYVE-1 expression is largely restricted to LECs. We therefore utilized a common marker together with LYVE-1 to distinguish between FRCs and LECs within the mLN. These stains revealed that these two stromal cell types were always present within close vicinity of one another, both in naive and infected mice (Fig. 2a, b). By staining thick (≥40 μm) vibratome sections with PDPN and LYVE-1 we could identify a mix of both sprouting and quiescent lymphatic vessels. Both PDPN and LYVE-1 expression was enriched on quiescent lymphatic vessels (Fig. 2b, *green arrows*), while the sprouting vessels showed a lower PDPN and LYVE-1 expression (Fig. 2b, *white arrows* and Supplementary movie 6). Furthermore, we analyzed FRCs (using a FRC specific maker ER-TR7[+]) and LECs (using LYVE-1[+]) location within interfollicular regions and observed a similarly close association between these two stromal populations (Fig. 2c). Finally we used a widely employed protocol utilizing PDPN and CD31 makers to distinguish FRCs (PDPN[+]CD31[−]) and LECs (PDPN[+]CD31[+]) by flow cytometry[26, 27]. This analysis supported our histological data and showed increased numbers of both FRCs and LECs following

*Hp* infection (Supplementary Fig. 2a–c). For both populations of cells, increases in number were driven, at least in part, by cell proliferation as determined using the proliferation marker Ki-67 (Supplementary Fig. 2a–c).

Taken together these data show that lymphatic vessel growth occurs in close association with the expansion of FRCs and raises the possibility that cross talk between LECs and FRCs may be occurring which in turn can govern the B-cell follicle organization.

**Lymphotoxin signaling to FRCs promotes lymphangiogenesis.** VEGF-A and VEGF-C promote lymphangiogenesis, and lymphotoxin-beta receptor (LTβR) signaling can elicit VEGF production by mesenchymal cells during LN anlagen formation[28]. FRCs have been reported to produce VEGF-A in naive LNs of mature mice[29] leading us to hypothesize that activation of FRCs by LTβR ligands may elicit VEGF-A and/or VEGF-C production and lymphatic cell growth. To address this hypothesis we crossed *Ccl19*$^{-cre}$ mice with *LTβR*$^{fl/fl}$ mice generating animals in which LTβR was selectively lost on *Ccl19*-expressing FRCs[30]. We utilized the same *Ccl19*$^{-cre}$ mice and crossed these to a line expressing enhanced yellow fluorescent protein (eYFP) from the *Rosa26* locus (*Rosa26*-eYFP) to confirm that Cre activity was restricted to FRCs. Analysis of mLNs from *Ccl19*$^{-cre}$ × *Rosa26*$^{-eYFP}$ mice confirmed expression of the *Ccl19*$^{-cre}$ transgene by FRCs, but not by LECs, BECs or on high endothelial venules (Supplementary Fig. 3a–d). In keeping with our hypothesis *Ccl19*$^{-cre}$ × *LTβR*$^{fl/fl}$ mice (*LTβR*$^{fl/fl}$) exhibited reduced lymphangiogenesis compared to control *Ccl19*$^{-cre}$ × *LTβR*$^{+/+}$ mice (*LTβR*$^{+/+}$) at both 12 and 21 dpi (Fig. 3a, b).

Lymphangiogenesis is also critical for promoting DCs entry to the draining LNs[3, 4]. Staining of mLNs from *LTβR*$^{fl/fl}$ and *LTβR*$^{+/+}$ mice for CD11c$^+$ DCs revealed that LEC expansion

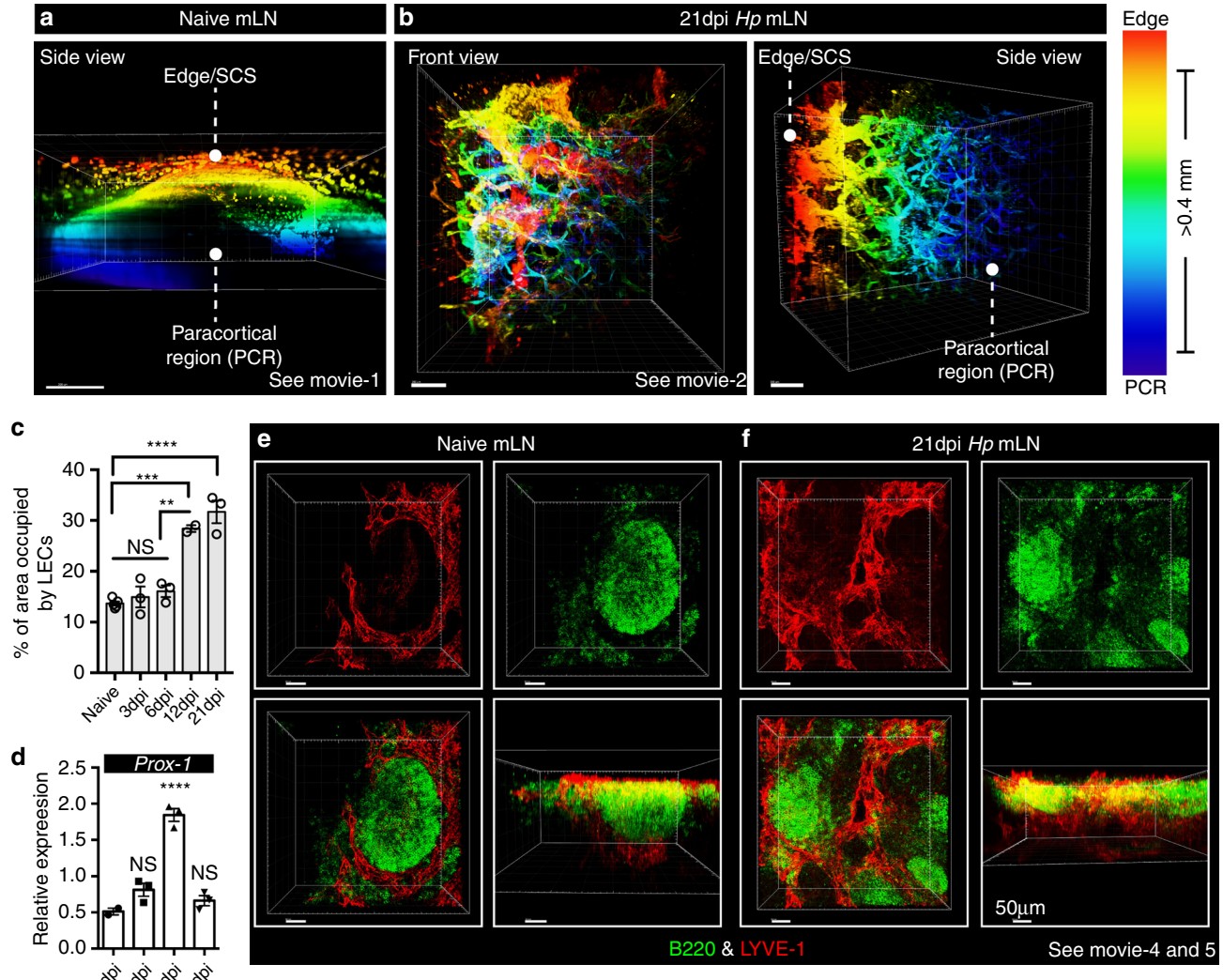

**Fig. 1** Intestinal helminth infection triggers mLN lymphangiogenesis. C57BL/6 J mice were infected with *Hp* and the entire chain of the mLN collected at the indicated time-points. **a**, **b** LYVE-1$^+$ lymphatic networks in the mLN of naive and 21 dpi infected mice were visualized by deep tissue imaging. Temporal color profiles are used to indicate the LYVE-1$^+$ lymphatic network in relation to cortical and paracortical regions. SCS; subcapsular sinus (*red*), *PCR* paracortical region (*blue*). **c** Quantitation of the mLN area occupied by lymphatics (LYVE-1$^+$ LECs) in C57BL/6 mice following *Hp* infection. Data are pooled from two independent experiments and represent mean ± SEM. Each *dot* represents an individual animal. **d** Relative expression of mRNA encoding Prox-1 in whole mLN over the indicated time course. Data is from a single experiment and is representative of ≥3 independent experiments. **e**, **f** Vibratome sections of mLNs showing 2D and 3D views of combined immunofluorescence staining for B-cell follicles (B220; *green*) and lymphatics (LYVE-1; *red*). *Scale bar* = 50 μm. Images are taken from a single mouse and are representative of three independent experiments. For all data shown, each independent experiment included n ≥ 2–3 mice/group/time-point. See also Supplementary Movies 1–2 and Supplementary movies 4–5

correlated with an increased number of DCs within the mLN of $LT\beta R^{+/+}$ mice, while mLNs from $LT\beta R^{fl/fl}$ mice harbored significantly fewer DCs (Fig. 3c, d). Overall these data reveal a crucial role for LTβR expression by FRCs in promoting infection-induced lymphangiogenesis and DCs accumulation in the mLN.

To determine which cell type was responsible for delivering the necessary lymphotoxin signals to FRCs we created mixed bone marrow chimeras mice (BMCs) in which B cells ($Jht^{-/-}$+ $LT\beta^{-/-}{\to}WT$) or T cells ($TCR\beta\delta^{-/-}$+$LT\beta^{-/-}{\to}WT$) failed to express lymphotoxin. Mixed BMCs using wildtype (WT) donors were also generated as controls ($Jht^{-/-}$+WT→WT and $TCR\beta\delta^{-/-}$+WT→WT). WT mice were used as recipients to ensure that the mice had a proper lymphoid structure, and normal stromal cell populations, prior to the irradiation. Naive mLNs of all chimeric

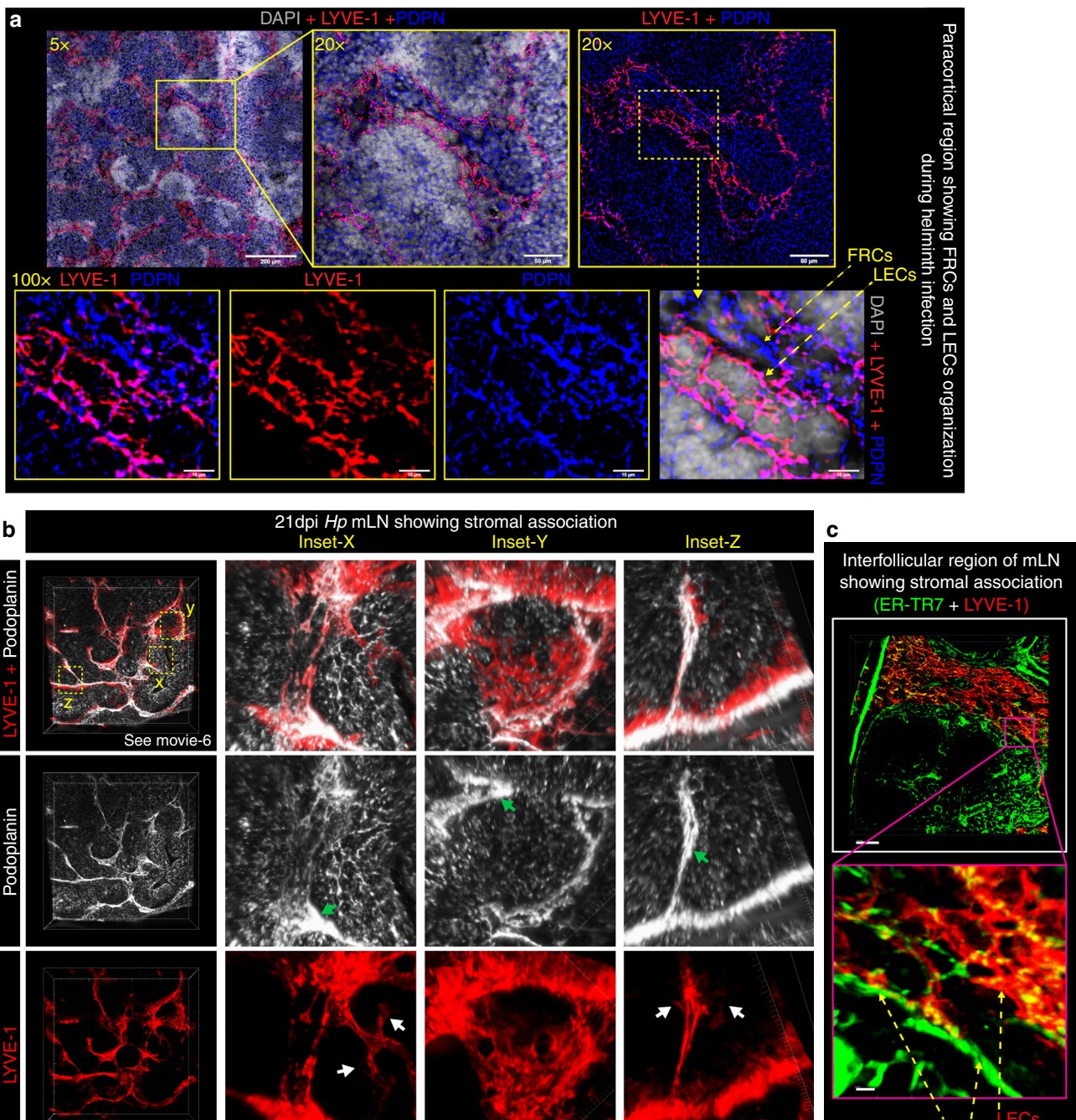

**Fig. 2** LECs and FRCs form intimate contacts in the area surrounding B-cell follicles. C57BL/6 J wild-type mice were infected with *Hp* and the entire chain of the mLN collected at 21 dpi. **a** 8 μm thick mLN cryosections showing combined immunofluorescence staining for FRCs (PDPN⁺; *blue*) and LECs (LYVE-1⁺; *red*) of *Hp*-infected mice are shown at various magnifications. Scale bar 50 μm. **b** Vibratome sections of ≥100 μm thick mLN showing 2D and 3D views of paracortical LECs (*red*; PDPN⁺LYVE-1⁺) and FRCs (*gray*; PDPN⁺LYVE-1⁻) lying in close vicinity of one another and surrounding a centrally located B-cell follicle. Insets (*x, y, z*) show higher magnification of FRCs and LECs. *Green* and *white* arrows highlight quiescent and sprouting lymphatic vessels respectively. **c** Vibratome sections of thick mLN section showing the mLN interfollicular region and highlighting associations between FRCs and LECs. All images are taken from a single mouse and are representative of two or more independent experiments, each including n ≥ 2 mice/group/time-point. See also Supplementary movie 6

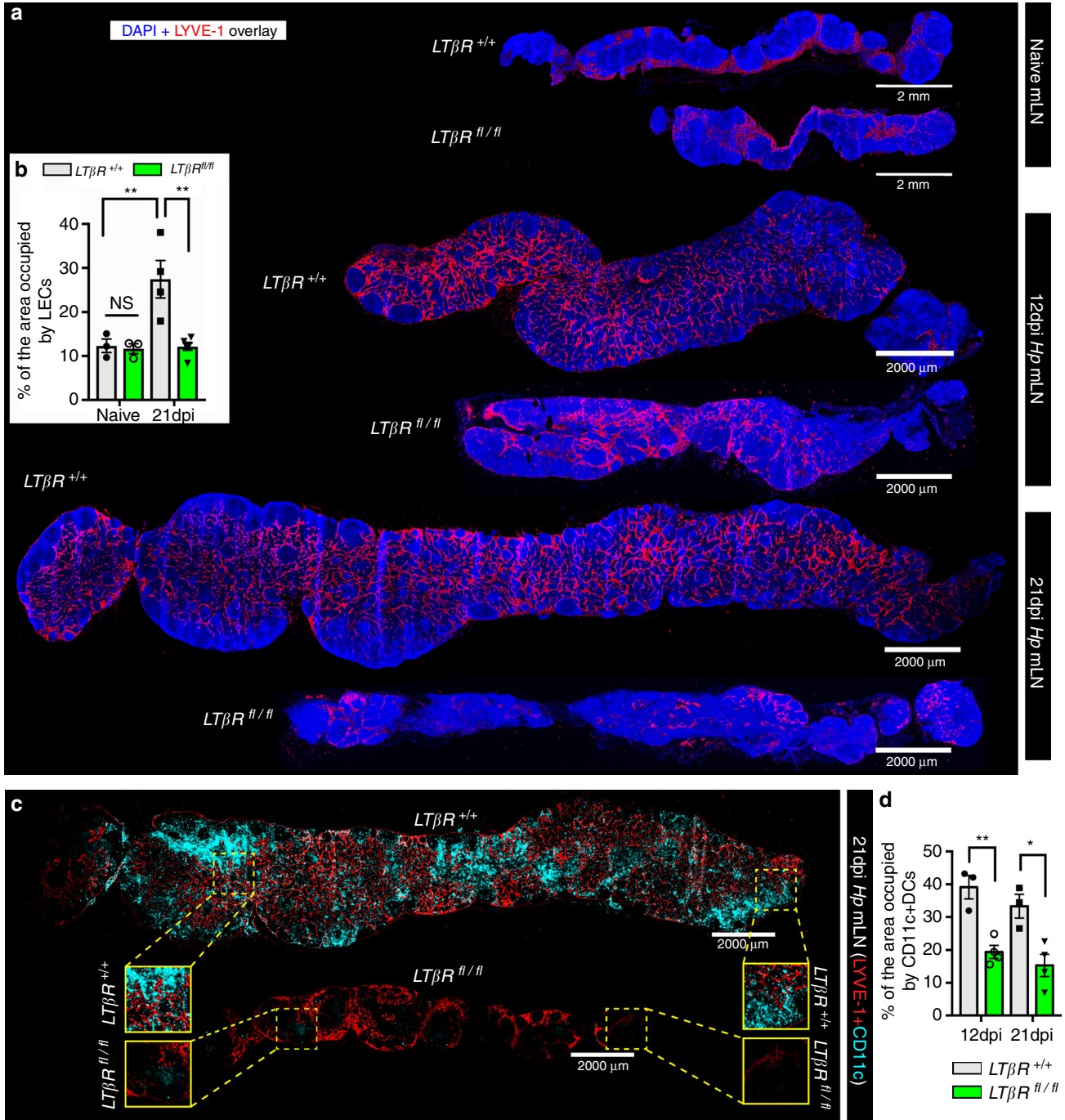

**Fig. 3** LTβR expression by *Ccl19*$^{-cre}$-positive stromal cells is required for mLN lymphangiogenesis. *Ccl19*$^{-cre}$ × *LTβR*$^{fl/fl}$ (*LTβR*$^{fl/fl}$) and wild-type littermate control *Ccl19*$^{-cre}$ × *LTβR*$^{+/+}$ (*LTβR*$^{+/+}$) mice were infected with *Hp* and the entire chain of the mLN collected at the indicated time-points. **a** mLN serial cryosections showing lymphatics in naive mice or at 12 and 21 dpi (*blue*; DAPI, *red*; LYVE-1$^+$). *Scale bar* = 2000 μm. Images are from single mice and are representative of two independent experiments each including *n* ≥ 2–3 mice/group/time point. **b** Quantitation of the mLN area occupied by LYVE-1$^+$ LECs in naive and *Hp*-infected animals. Data represents mean ± SEM and are pooled from two independent experiments with each *dot* representing an individual mouse. **c** mLN serial cryosections showing LYVE-1$^+$ stromal cell organization and DC distribution within the mLN at 21 dpi (*Cyan*; CD11c and *red*; LYVE-1$^+$). *Scale bar* = 2000 μm. **d** Quantitation of the mLN area occupied by DCs (CD11c$^+$) in infected *LTβR*$^{+/+}$ and *LTβR*$^{fl/fl}$ mice. Data represents mean ± SEM and are pooled from two independent experiments with each dot representing an individual mouse. Statistical analyses were performed using ANOVA, Bonferroni's multiple comparison test and significance donated as *$P$ < 0.05, **$P$ < 0.01, ***$P$ < 0.001, and ****$P$ < 0.0001

mice showed a normal lymphatic network that was restricted to the area below the SCS and the medullary regions (Supplementary Fig. 4a–d). The absence of lymphotoxin on B cells but not T cells, resulted in reduced LN swelling and lymphangiogenesis in response to *Hp* infection (Fig. 4a, d). Of note, naive *CCL19*$^{cre}$ × *LTβR*$^{fl/fl}$ mice and *Jht*$^{-/-}$+*LTβ*$^{-/-}$→WT BMCs exhibited normal numbers of B-cell follicles in the naive state, but failed to develop de novo B-cell follicles in response to infection[19]. This indicates the possible existence of an LTβR-dependent positive feedback loop between B cells and FRCs that promotes the expansion of both cell types, and which further supports the proliferation of LECs.

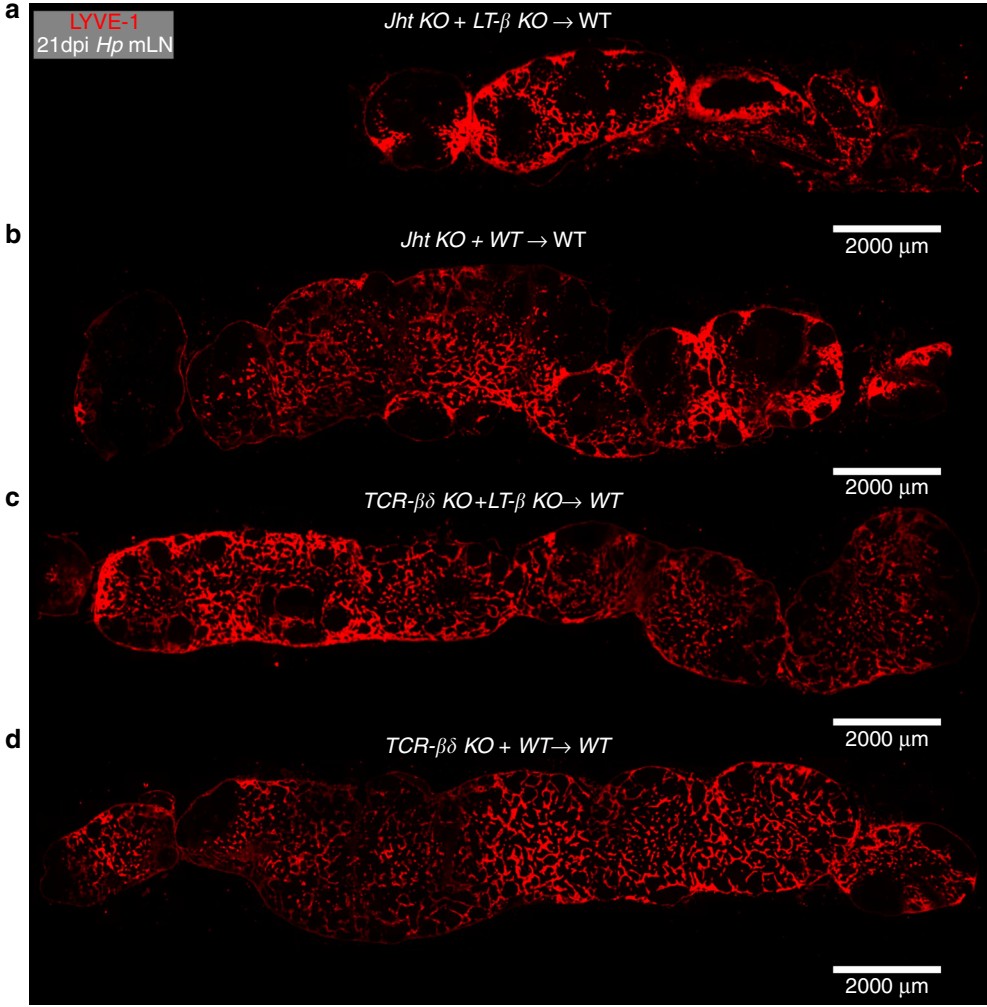

**Fig. 4** B cells provide the lymphotoxin required to drive mLN lymphangiogenesis. Mixed bone marrow chimeras were generated as described in Methods section. Chimeras lacking lymphotoxin expression exclusively on B cells ($Jht^{-/-}+LT\beta^{-/-}$) or T cells ($TCR\delta\beta^{-/-}+LT\beta^{-/-}$) were compared to control mice receiving B-cell or T-cell deficient bone marrow mixed with WT cells ($Jht^{-/-}+WT$ and $TCR\delta\beta^{-/-}+WT$). All mice were infected with *Hp* and the entire chain of the mLN collected at day 0 (naive) and 21 dpi. **a, b** mLN cryosections from infected chimeric mice lacking lymphotoxin on B cells ($Jht^{-/-}+LT\beta^{-/-}$) and their respective controls ($Jht^{-/-}+WT$) showing immunofluorescence images staining for LYVE-1[+] (*red*) LECs network. **c, d** mLN cryosections from mixed BMC infected mice lacking lymphotoxin on T cells ($TCR\delta\beta^{-/-}+LT\beta^{-/-}$) and respective controls ($TCR\delta\beta^{-/-}+WT$) showing immunofluorescence images staining for LYVE-1[+] (*red*) LECs network. *Scale bar* = 2000 μm

**B cell–FRC cross talk promote VEGF-A and VEGF-C production**. We next assessed whether activated FRCs produced increased VEGF-A and/or VEGF-C to promote lymphangiogenesis in *Hp*-infected mice. To our surprise examination of *Vegf-a* and *Vegf-c* mRNA expression in stromal vs. hematopoietic compartments of the mLN revealed that increased transcripts could only be detected within the hematopoietic compartment (Supplementary Fig. 5a). FACS analysis confirmed the CD45[+] hematopoietic compartment as the major source of VEGF-A and VEGF-C producing cells following helminth infection (Supplementary Fig. 5b, c, *blue histograms*). FACS analysis revealed that B cells were the main source of both VEGF-A and VEGF-C (Supplementary Fig. 6a–d). Both naive (IgD[+]) and antigen-activated (IgD[−]) B cells produced VEGF-A and VEGF-C in response to helminth infection, however, production was strongly enriched within the antigen-activated B cells (Supplementary Fig. 6c, d). Of interest, activated B cells also represented the main source of increased lymphotoxin expression following helminth infection (Supplementary Fig. 6e). These data indicated that activated B cells, but not FRCs, were responsible for increased VEGF production following *Hp* infection despite the requirement for lymphotoxin-dependent activation of FRCs in promoting lymphangiogenesis. We therefore reasoned that LTβR expressing FRCs likely promote lymphangiogenesis by providing signals to interacting lymphotoxin-expressing B cells, that in turn function to elicit VEGF-A and VEGF-C production by these cells.

To dissect the molecular cues responsible for FRC–B cell cross talk we assessed the production of factors by lymphotoxin-activated FRCs that may result in altered production of VEGF-A and VEGF-C by activated B cells. FRCs located in the B-cell follicle mantle have previously been shown to produce B-cell-activating factor (BAFF) to promote B-cell survival[31]. Examination of *Hp*-infected mLNs revealed increased BAFF expression by ER-TR7[+]laminin[+] FRCs located within in the follicle mantle of both naive and infected mLN, but not within the follicular DC rich B-cell follicle (Fig. 5a–c and Supplementary Fig. 7a, b). Importantly an increased number and density of BAFF expressing FRCs could be observed in the mLN of *Hp*-infected mice, a finding that was supported by the observation that *Hp* infection increased BAFF mRNA expression in the stromal, but not the hematopoietic, compartment of the mLN (Supplementary Fig. 7a).

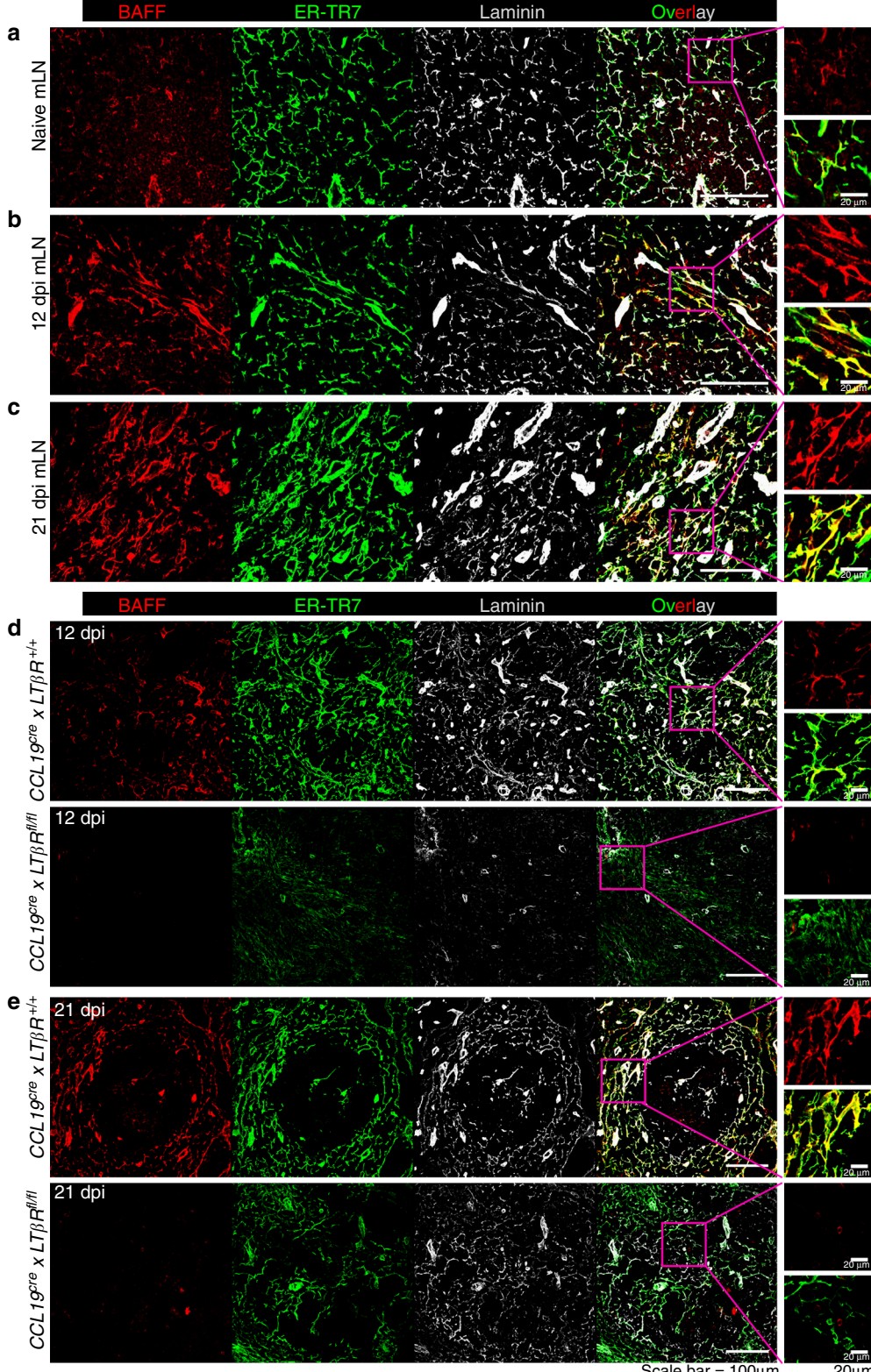

**Fig. 5** Intestinal helminth infection activates FRCs surrounding B-cell follicles to produce BAFF in a LTβR-dependent manner. C57BL/6 J wild-type mice were infected with *Hp* and the entire chain of the mLN collected at the indicated time-points and processed for histological staining. **a** Naive, **b** 12 dpi and **c** 21 dpi *Hp* mLNs showing BAFF (stained with anti-BAFF antibody) expression by FRCs (ER-TR7[+] laminin[+]). BAFF expression in mLN FRCs in *Hp*-infected *Ccl19[cre] × LTβR[fl/fl]* and *Ccl19[cre] × LTβR[+/+]* mice at **d** 12 dpi and **e** 21 dpi *Hp* infection. *Scale bar* 100 μm and 20 μm. The *magenta* inset in the overlay panel is amplified and shown for BAFF alone (*red*) or for ERTR-7[+] FRCs (*red+green*) expressing BAFF

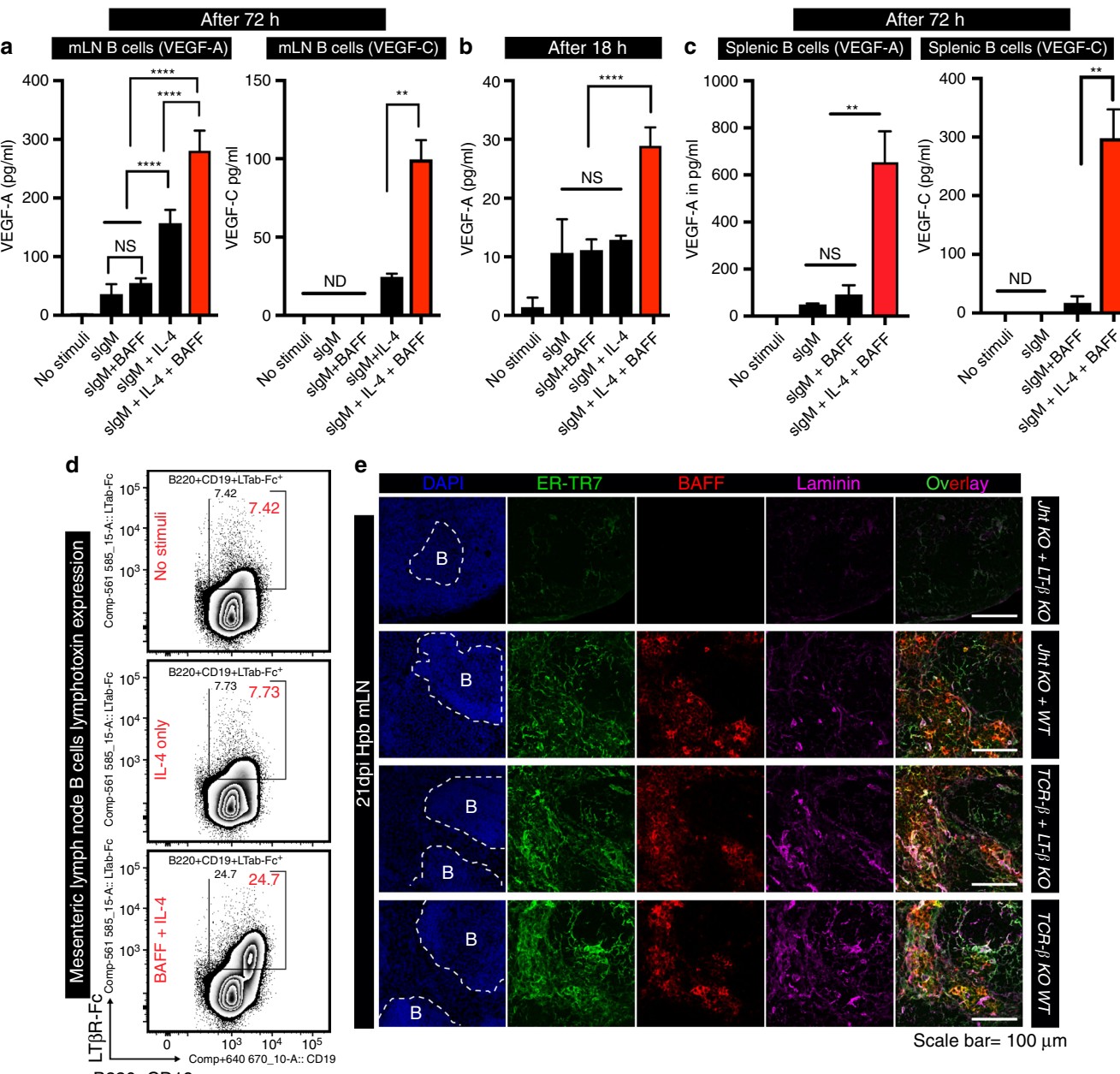

**Fig. 6** BAFF and IL-4 stimulation of B cells promotes lymphotoxin and VEGFs production. **a–c** mLN B cells were cultured with anti-IgM with or without IL-4±BAFF and the culture supernatant collected at **a** 72 h or **b** 18 h post stimulation for quantitation of VEGF-A and VEGF-C by ELISA. **c** Splenic B cells were stimulated as in (**a**) and the culture supernatant collected at 72 h for quantitation of VEGF-A and VEGF-C by ELISA. Data represent mean ± SEM, and are representative of pooled data from three independent experiments **a** or are representative of two independent experiments **b**, **c**. **d** Naive mLN B cells were cultured with or without IL-4±BAFF for 18 h then B-cell lymphotoxin expression determined by staining with LTβR-Fc. Data represent mean ± SEM and representative of three independent experiments. **e** mLN cryosections from mixed BMC-lacking lymphotoxin on B or T cells and infected with *Hp* for 21 days were stained for BAFF⁺ (*red*) and FRCs (ER-TR7⁺Laminin⁺). *Scale bar* = 100 μm. Statistical analyses were performed using ANOVA, Bonferroni's multiple comparison test and significance donated as *$P < 0.05$, **$P < 0.01$, ***$P < 0.001$, ****$P < 0.0001$; *NS* not significant and *ND* not detectable

FRCs required activation via LTβR to produce BAFF, as BAFF protein expression was ablated in *Hp*-infected *LTβR^{fl/fl}* mice at both 12 and 21 dpi (Fig. 5d, e).

BAFF is a known survival factor for B cells and can additionally enhance B-cell chemotaxis towards CXCL-13[32, 33]. However, a role for BAFF in promoting lymphangiogenesis has not been reported. To determine whether BAFF could impact on activated B cells to elicit VEGF production we stimulated naive mLN B cells in vitro with anti-IgM antibodies together with IL-4 or BAFF plus IL-4. BAFF synergized with IL-4 to promote VEGF-A

and VEGF-C production by B cells stimulated with anti-IgM for 72 h (Fig. 6a). Importantly this effect was not due to altered B-cell survival as the ability of BAFF, in combination with IL-4, to elicit VEGF production was also apparent by 18 h after stimulation, a timepoint at which no impact of BAFF on B-cell viability could be observed (Fig. 6b). It was also not limited to B cells isolated from the mLN as BAFF and IL-4 also stimulated VEGF-A and VEGF-C production by activated splenic B cells (Fig. 6c). Of note addition of BAFF and IL-4 to anti-IgM-activated B cells also resulted in the upregulation of lymphotoxin by these cells

(Fig. 6d), and activation of FRCs by lymphotoxin-expressing B cells is required for BAFF expression (Fig. 6e).

**Neutralizing anti-BAFF mAb attenuates mLN lymphangiogenesis.** To further validate the role of BAFF in promoting mLN lymphangiogenesis we performed in vivo experiments in which BAFF

was neutralized using the anti-BAFF mAb, (Sandy-2)[34]. Animals were treated with 2 mg/kg of the Sandy-2 or istoype control mAb by intraperitoneal injection just prior to and 5 days following Hp infection (Fig. 7a). To ensure that BAFF blockade was achieved we analyzed B-cell populations in the inguinal LN (iLN) of naive animals treated with the anti-BAFF mAb. As expected based on

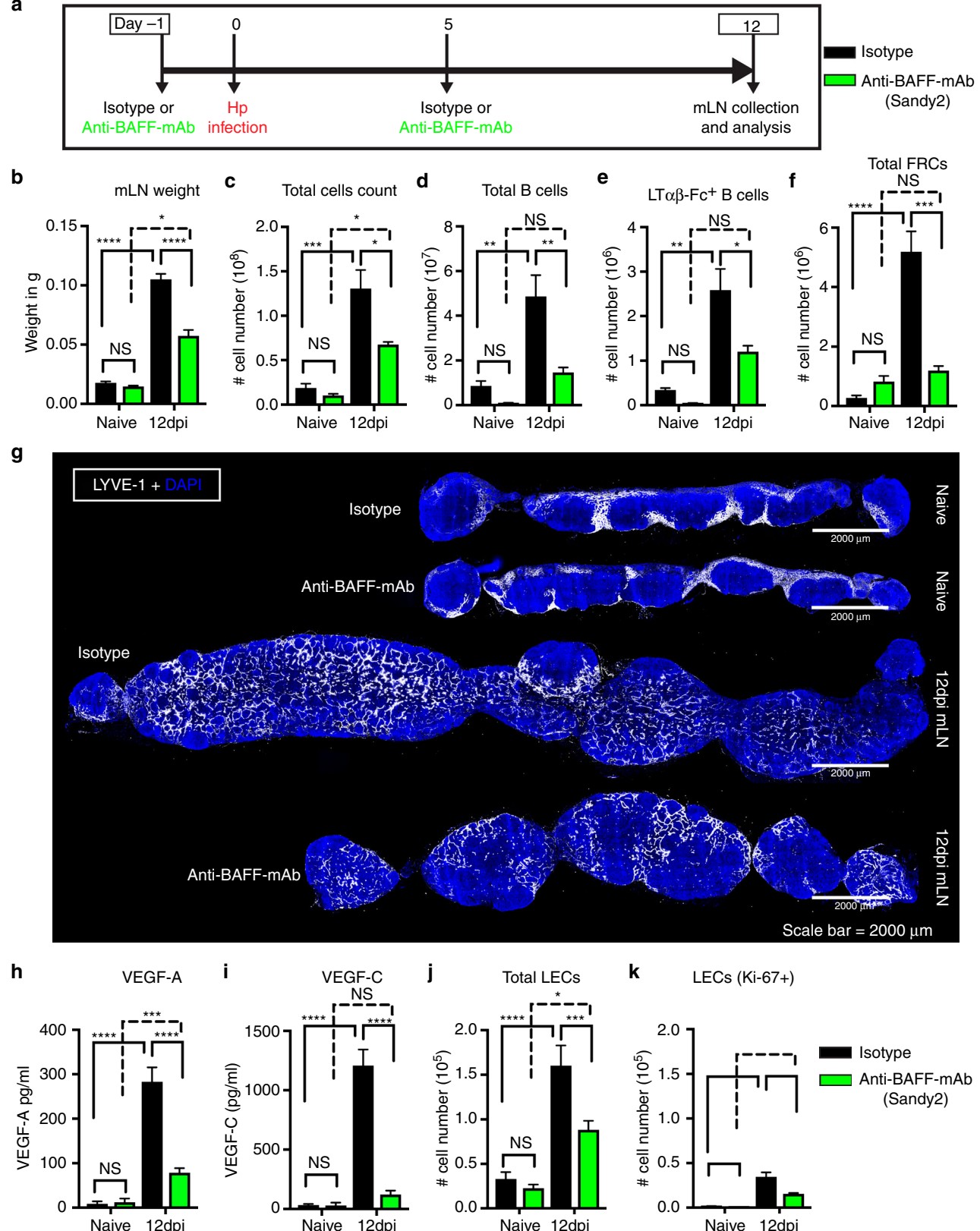

previous publications[34] the iLN of mice treated with anti-BAFF mAb harbored significantly decreased numbers of B cells (Supplementary Fig. 8a). Treatment of *Hp*-infected mice resulted in a significant reduction in the weight, cellularity, and B-cell number (Fig. 7b–d, Supplementary Fig. 8b). There was also a significant reduction of lymphotoxin-expressing B cells, and an associated reduction in the total number and proliferation of FRCs (Fig. 7e, f, Supplementary Fig. 8c–e). Finally, we observed reduced lymphangiogenesis (Fig. 7g) and reduced production of both VEGF-A and VEGF-C (Fig. 7g–i) along with reduced total and proliferating LECs (Fig. 7j, k) in the mLN of mice treated with anti-BAFF mAb. These data confirm a role for BAFF in promoting VEGF production and lymphatic growth in response to helminth infection.

In summary, our data indicates that BAFF provides a feed-forward loop in which B-cell-mediated FRC activation results in the upregulation of lymphotoxin by B cells, thereby licensing these cells to provide further stimulatory signals to FRCs and to promote increased BAFF production (Fig. 8). These data likely explain the requirement for B cell–FRC cross talk in eliciting increased VEGF production and promoting mLN lymphangiogenesis following *Hp* infection (Fig. 8).

## Discussion

During inflammatory responses the draining LNs swell to allow the recruitment and accumulation of increased numbers of lymphocytes, thereby increasing the probability of antigen-specific T and B cells meeting their cognate antigen. Stromal cells also expand to accommodate incoming lymphocytes, with FRCs and vascular cells (lymphatic and blood endothelial cells)

typically exhibiting a tightly coordinated process of growth[19, 35, 36]. Increased blood flow allows a continuous delivery of blood-borne lymphocytes, oxygen and nutrients, while an expanded network of lymphatic vessels promotes the delivery of antigens and stimulatory DCs from infected tissues.

In the current study we demonstrated that intestinal helminth infection, which elicits a strong type 2 response, drives the profound growth of lymphatic vessels within the draining mLN. We further investigated the cellular mechanisms underlying mLN lymphangiogenesis in helminth-infected mice and report a previously unrecognized role for lymphotoxin-dependent FRC–B cell interactions in promoting mLN LEC expansion. Ligation of LTβR on FRCs by antigen-activated, lymphotoxin-positive, B cells promoted BAFF production by FRCs. BAFF in turn acted on B cells to elicit the production of the lymphangiogenic factors VEGF-A and VEGF-C, and to further upregulate lymphotoxin expression. These data unveil a complex interplay between FRCs, LECs and B cells that regulates the coordinated growth of a dense FRC–LEC network surrounding B-cell follicles. It is likely that this stromal cell network functions both to "physically" support the B-cell follicle in addition to providing signals, such as BAFF, that support B-cell survival. Previous studies have shown that FRCs are capable of producing VEGF[29, 36]. We also observed VEGF production by stromal cells in naive mice; however, B cells not stromal cells were found to represent the major source of increased VEGF production following helminth infection. Thus the presence or absence of inflammation, and also the exact nature of the inflammatory signal, likely determines which lymphoid cells represent the major source of VEGF. Interestingly, FRCs were shown to be sensitive to lymph flow[37], and inflammation-induced increases in lymph flow from peripheral

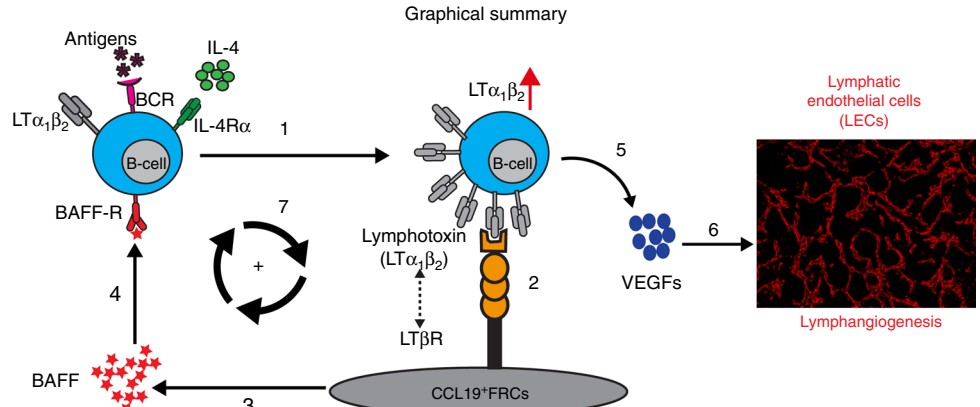

Graphical summary

**Fig. 8** Schematic of FRC–B cell interactions resulting in mLN lymphangiogenesis following *Hp* infection. Lymphotoxin expression by IL-4- and antigen-activated B cells (1) signals to LTβR expressed by FRCs (2) to promote the expression of BAFF (3). BAFF then synergizes with IL-4 to further upregulate lymphotoxin expression on antigen-activated B cells in a feed-forward manner that acts to amplify total BAFF levels (4). BAFF and IL-4 then synergize to promote VEGF production by B cells (5) which drives helminth-induced lymphangiogenesis and continued DC accumulation within the mLN (6). FRC-derived BAFF also stimulates lymphotoxin expression by interacting B cells creating a feed-forward loop that furhter promotes FRC activation (7)

**Fig. 7** In vivo BAFF inhibition attenuates helminth-induced lymphangiogenesis. **a** C57BL/6 J wild-type mice were treated with isotype control or anti-BAFF mAb (Sandy-2) and infected with *Hp*. The entire chain of the mLN was collected at 12 dpi and processed for flow cytometry or histological staining. **b** Total weight of mLN, **c** total cell count, **d** total number of B cells, **e** absolute number of lymphotoxin-expressing B cells, and **f** total FRCs (PDPN$^+$MadCAM1$^-$CD31$^-$) present within the mLN as determined using flow cytometry. **g** mLN serial cryosections showing lymphatic organization after treatment with isotype control or anti-BAFF mAbs in naive or at 12 dpi mice (blue; DAPI, grays; LYVE-1$^+$ LECs). Scale bar = 2000 μm. Images are from a single mouse and are representative of two independent experiments each including n ≥ 2 mice/group/time point. **h** VEGF-A and **i** VEGF-C in mLN tissue homogenates as determined by ELISA. **j, k** total LECs (PDPN$^+$CD-31$^+$) and proliferating LECs (PDPN$^+$CD-31$^+$Ki-67$^+$) in the mLN of naive and 12 dpi mice treated with isotype control or anti-BAFF mAbs were determined using flow cytometry. Data represent mean ± SEM and representative of two independent experiments with n = 3 mice/group/time-point. Statistical analyses were performed using ANOVA, Bonferroni's multiple comparison test and significance donated as *P < 0.05, **P < 0.01, ***P < 0.001, and ****P < 0.0001

tissues to the draining lymphoid organs have been shown to influence VEGF production by FRCs[36]. Thus it is plausible that the small increase in VEGF-A production and LEC numbers observed in mice treated with anti-BAFF mAb may result from increased lymph flow elicited by helminth infection[36]. It is also possible that the FRC–LEC network surrounding B-cell follicles supports B-cell functions in other ways that are yet to be determined. In this regard it is already known that BAFF promotes efficient germinal center responses and isotype class switching[24, 38]. This is in keeping with our previous observations that Hp-infected LTβR[fl/fl] mice exhibit reduced antibody production, and harbor increased adult worms[19]. Thus it would be interesting to investigate the impact of FRC-derived BAFF on fostering B-cell follicle formation, germinal center reactions, and antibody production in response to Hp infection.

LECs can also express LTβR[27, 39, 40] and so may communicate directly with lymphotoxin-expressing B cells. However, the dramatic loss of lymphatic growth in mice lacking LTβR specifically on FRCs indicates that a possible contribution of B cell–LEC cross talk to lymphangiogenesis must be secondary to B cell–FRC communication. Interestingly Furtado et al.[41], reported that lymphotoxin fails to elicit the proliferation of LECs in vitro, but does promote LEC tube formation. Thus, while lymphotoxin-dependent B cell–FRC interactions function to promote VEGF production and drive LEC proliferation, B cell–LEC interactions may function to ensure proper tube formation.

IL-4 is a cytokine with pleiotropic activity in the immune system and serves as an important factor for driving protective immune response against helminth infection[23]. Recent studies have shown a negative regulatory role for IL-4Rα signaling in lymphatic growth[17, 42], likely through a direct action of IL-4/IL-13 on LECs. Paradoxically, type 2 driven inflammation associated with allergic disease is typically associated with increased lymphatic growth, and inflammatory lymphangiogenesis was reported within the dermis of mice expressing IL-4 under the control of a keratinocyte-specific promoter[43]. Zhang et al.[44] also reported that activation of human RAW264.7 macrophages with IL-4, but not by IFN-γ+LPS, elicited VEGF production and promoted LEC growth and tube formation both in vitro and in vivo in a mouse model of Lewis lung carcinoma. Thus, IL-4–IL-4Rα signaling may play a dual function in lymphangiogenesis, on the one hand promoting lymphangiogenesis by upregulating lymphotoxin and VEGF production on macrophages and B cells, and on the other hand by acting directly on LECs to limit proliferation. Future studies with IL-4Rα deficient animals will be needed to determine whether the action of type 2 cytokines on lymphatic vessels differs in distinct tissues or during acute vs. chronic type 2 inflammation.

A role for B cells in providing VEGF-A to support inflammatory lymphangiogenesis has been reported following LPS delivery[13] or vaccination of mice with the model antigen keyhole limpet hemocyanin emulsified in complete Freund's adjuvant[3]. Thus, it will be interesting to determine whether B cell–FRC interactions are also important in supporting lymphatic vessel growth in this, or other inflammatory settings. It is also clear that other forms of inflammation can elicit lymphoid lymphangiogenesis and further exploration of the role of other inflammatory cytokines or stimuli in regulating B cell–FRC cross talk is warranted. In the context of viral and bacterial infections the impact of pathogen-associated molecular patterns might be of particular relevance.

Lastly, it would be interesting to explore the impact of intranodal lymphangiogenesis on functions beyond promoting DC entry. Another important function may be to promote the exit of effector T cells and plasmablasts from the LN[45]. This step is crucial for the subsequent entry of effector T cells into the

inflamed intestinal lamina propria or for plasmablasts into the bone marrow. In the context of Hp infection, antibody production is particularly important for protecting the host against repeated infections, thus the impact of intranodal lymphangiogenesis on the migration of plasmablasts to the bone marrow and on long-lived antibody responses would be particularly relevant. In this regard the remodeling of stromal networks so that lymphatic vessels arise adjacent to the newly developed B-cell follicles present in Hp-infected mice makes sense in that this would reduce the distance necessary for plasmablasts exiting the germinal center and migrating to the bone marrow[46].

In summary we have unveiled a critical role for lymphotoxin-dependent B cell–FRC cross talk in regulating lymphatic growth and DC entry into the inflamed mLN following helminth infection. These data highlight the complexity of cellular interactions within the inflamed LN and provide new insight into how the regulation of intranodal lymphangiogenesis occurs.

## Methods

**Ethics statement**. All animal experiments were approved by the Service de la consommation et des affaires vétérinaires (1066 Épalinges, Canton of Vaud, Switzerland) with the authorization numbers VD 2238.1 and VD 3001.

**Mice strains and parasites and treatments**. C57BL/6 J (WT) mice were obtained from Charles River and randomly distributed by animal caretaker before start of any experiment. Ccl19[−cre] × LTβR[+/+] (LTβR[+/+]) and Ccl19[−cre] × LTβR[fl/fl] (LTβR[fl/fl]) were provided by Kantonsspital St. Gallen and were maintained on the C57BL/6 J background under specific pathogen-free (SPF) conditions at École Polytechnique Fédérale de Lausanne (EPFL), Switzerland. LT-β[−/−] and TCRβδ[−/−] mice (C57BL/6 J background) were maintained at Epalinges animal facility, University of Lausanne, Switzerland. During the course of study all the mice were infected orally with 200 L3 stage Hp and mice sacrificed at indicated time-points post infection.

**Flow cytometry and antibodies**. At given experimental time point, mice were killed and mLN were isolated for flow cytometry. Single-cell suspensions were prepared using enzyme mixture comprised of RPMI-1640 medium containing 0.8 mg/ml Dispase, 0.2 mg/ml Collagenase P (both from Roche), and 0.1 mg/ml DNase I (Invitrogen). mLN were incubated at 37 °C and gently mixed using a pipette at 5 to 15 min intervals to ensure the proper dissociation. After complete dissociation mLN cells were filtered through a 40-μm cell strainer, counted, and used for surface staining. For lymphocyte staining mLN single-cell suspensions were gently dispersed, cells filtered through 40-μm-cell strainer, counted and used for FACS surface staining. For stromal cell staining mLN were subjected to enzymatic digestion using a digestion mixture comprised of RPMI-1640 containing Dispase and Collagenase P (both from Roche), and 0.1 mg/ml DNase I (Invitrogen). Then cells were resuspended in FACS buffer (PBS containing 2% FBS and 5 mM EDTA). For staining, cells were incubated for 30 min with antibodies against the indicated markers and the samples acquired on BD-LSRII machine and data were analyzed using FlowJo (v10.0.6). The CD45-negative fractions that were positive for podoplanin (PDPN) and CD31 were identified as LECs, PDPN[+], and CD31[−] were FRCs[26, 27]. The proliferating populations were identified as CD45[−]PDPN[+]CD31[+]Ki-67[+] as LECs and PDPN[+]CD31[−]Ki-67[+] were FRCs. For detection of VEGFs expression in CD45[+] and CD45[−] cells, surface staining were first performed using rat anti-mouse CD45 antibody (BioLegend) followed by intracellular staining using transcription factor staining buffer set (eBiosciences, Cat. No. 00-5523-00). For VEGF-A and VEGF-C, cells were permeabilize, and stained using rabbit anti-mouse–VEGF-A or C antibody and revealed with anti rabbit secondary antibody coupled to alexa-568. A rabbit IgG control antibody (R&D systems) was used as an isotype control. The samples were acquired on BD-LSRII machine and data were analyzed using FlowJo (v10.0.6). Lymphotoxin-expressing B cells were identified using LTβR-Fc staining as described bellow. A detailed list of antibodies used for flow cytometry is provided in Supplementary table 1.

**Histology and immunofluorescence microscopy**. The entire length of the mLN chain was carefully dissected, weighed, imaged, and embedded in Tissue-Tek optimum-cutting temperature compound (Thermo Scientific), then frozen in an iso-pentane dry ice bath. Serial cryostat sections (8 μm in thickness) were collected over a span of 400 μm depth on Superfrost/Plus glass slides (Fisher Scientific), air-dried and fixed for 10–15 min in ice-cold acetone. Air-dried cryosections were then rehydrated in PBS and were blocked with 1% (wt/vol) BSA supplemented with normal mouse (1%) and donkey serum (4%). Indirect immunofluorescence staining was performed using various antibodies (listed in Supplementary table 2) diluted in PBS containing 1% (wt/vol) BSA and 1% (vol/vol) normal mouse serum. Cryosections were incubated with primary antibodies overnight at 4 °C. After overnight incubation cryosections were washed three times in PBS and primary

antibodies were detected by incubating sections with fluorescently labeled secondary antibodies, and nuclei counter-stained with DAPI prior to mounting of the sections using ProLong anti-fade reagents (Life technologies). Stained cryosections were then imaged after 24 h. A detailed list of antibodies used is provided in Supplementary table 2.

**Vibratome sections.** Isolated mLN were fixed overnight at 4 °C in freshly prepared 1–2% paraformaldehyde in PBS, washed, and embedded in 2% (w/v) low-melting agarose (Sigma-Aldrich) in PBS. 200 to 500-µm thick sections were cut with a vibratome (Microm HM 650 V) and were used for staining. These thick sections were blocked with blocking buffer (as described above) overnight and stained for at 2 to 5 days with the primary antibodies followed by extensive washing in PBS (total 5×/ 1 h each) before incubation with fluorescently labeled secondary antibodies. After staining, samples were cleared described previously[19]. After clearing, vibratome sections were imaged using light sheet microscope (Zeiss) with 20×objective in a 80.2% fructose solution. The 3D reconstruction and movies were made using IMARIS (Bitplane). In separate set of experiments, the whole mLN was cleared using X-CLARITY Electrophoretic Tissue Clearing System (Logos) and then processed for sectioning and immunofluorescence staining for 5 to 10 days before imaging on a light sheet microscope. The 3D reconstruction and movies were made using IMARIS.

**Image acquisition and processing.** Images were acquired on an Olympus VS120-SL full slide scanner using a 20×/0.75 air objective and an OlympusXM10 B/W camera or LSM710 laser scanning confocal microscope or with light sheet microscope (Zeiss) with 20× objective. For images that were acquired using Olympus VS120-SL full slide scanner or with LSM710 laser scanning confocal microscope, each image was acquired using the indicated fluorescent channels and the same exposure time employed across different samples. The images were down sampled when extracted using the VSI reader action bar developed by the EPFL BioImaging & Optics Platform (BIOP) and were then subjected to the analysis pipeline available through ImageJ/Fiji. For generation of the final images comparing different samples (i.e., naive vs. infected mLN), each fluorescent channel was set to the same brightness and contrast, the mLN chain outlined using Fiji, then assembled such that the final image represented the individual and overlay of all channels. Alternatively, images were directly processed using Olympus slide scanner software (OlyVIA v.2.6) after adjusting the brightness and contrast settings so that they remained the same across all samples compared. For quantitative measurements, immunofluorescence images from naive and infected mice mLN were acquired and segmented using ImageJ software and the number of pixels specific for given marker against DAPI was measured using an automated macro and expressed as the percentage of total pixels in each area occupied by given marker.

**Bone marrow chimeras.** All mice were maintained in specific pathogen-free conditions. For the generation of B cells or T cells lacking lymphotoxin expression (B-$Lt\beta^{-/-}$ or T-$Lt\beta^{-/-}$) C57BL/6 J recipients were reconstituted with 80% $Jht^{-/-}$ or LSM710 laser scanning confocal $Lt\beta^{-/-}$ bone marrow. Controls were generated in which C57BL/6 J (WT) recipients were reconstituted with 80% $JhT^{-/-}$ or 80% $TCR\beta\delta^{-/-}$ bone marrow plus 20% WT bone marrow, respectively. All recipient mice received the antibiotic "Baytril 10%" (1/1000) in the drinking water for 4 to 8 weeks following bone marrow reconstitution and where subjected to infection at 8 weeks following reconstitution.

**In vitro B-cell culture and LTβR-Fc staining.** Single-cell suspensions from the naive C57BL6/J mice mLN or spleen were prepared in complete RPMI-1640 medium (RPMI+10%FCS+penicillin/streptomycin+glutamine+HEPES). B cells were separated by negative selection on a magnetic column, according to the manufacturer's instructions (Miltenyi Biotech). Freshly purified mLN B cells were then cultured in complete RPMI-1640 medium overnight before stimulation. After overnight resting cells were stimulated in the presence of recombinant mouse BAFF (R&D systems, 2106-BF-010/CF) with and with out rIL-4 (PeproTech). After the indicated time point, cultured cells were washed twice with PBS and stained for surface LTβR-Fc as described previously[19, 47]. In brief, Single-cell suspensions were prepared from the mLN and were subjected to surface blocking using FACS buffer (PBS+5 mM EDTA+0.1% azide+2%FCS) containing anti-CD16/32; 2.4G2+0.5% normal mouse serum and normal rat serum. After 20 min, 25 µl of LTβR-Fc (Biogen 1 mg/ml; diluted 1:50) were added on top in FACS buffer. After 30 min of incubation, cells were washed thrice with FACS buffer and 25 µl of biotinylated goat anti human IgG antibody (diluted 1:400, pre-absorbed for 30 min on ice with 4% mouse and 4% rat serum) was added. After 30 min cells were washed thrice with FACS buffer and stained using antibody cocktail containing fluorescent-coupled streptavidin as well as antibody to other surface marker. After 30 min cells were washed again and resuspended in 200 µl of FACS buffer and analyzed using flow cytometer.

**Analysis of VEGF secretion by B cells.** Purified B cells were stimulated with sIgM (AffiniPure F(ab')₂ fragment goat anti-mouse IgM, Jackson ImmunoResearch) with or without additional BAFF (100 ng/ml) and rIL-4 and culture supernatants

collected at indicated time point and stored at ≥−20 °C until used for VEGF-A or C protein analysis. In a separate set of experiments, 1 mg of mice mLN were homogenized in 1 ml of lysis buffer supplemented with complete protease inhibitors (Roche) and stored at −80 °C until assayed for VEGFs level using ELISA kits. VEGF-A levels in the culture supernatant or in tissue homogenates were determined using mouse VEGF quantikine ELISA kit (R&D system, Cat No. MMV00), where as VEGF-C levels were determined using mouse VEGF-C ELISA kit (CUSABIO, CSB-E07361m) according to the manufacturer's instructions. In another set of experiments, mice were sacrificed at indicated time point post infection and intracellular VEGFs expressing B cells were measured as described previously with out any further manipulations[36].

**In vivo BAFF inhibition.** For in vivo BAFF inhibition experiments anti-mBAFF monoclonal antibody (Sandy-2) (Adipogen, AG-20B-0063PF) or an isotype control antibody (BioXCell) was administered i.p. to 8 weeks old female C57BL6/j (WT) mice at 2 mg/kg on days −1 and 5 (Fig. 7a). During the course of study all the mice were either infected orally with 200 L3 stage *Hp* or left as naive control. Mice were sacrificed at indicated time-points post infection for the analysis of mLN. iLN were also collected and analyzed for B-cell and T-cell population to confirm the antibody effect as described previously[34].

**RNA isolation and qRT-PCR analysis.** The complete length of mLN were collected into trizol and stored at −80 °C until used. Stromal and cellular fraction were separated as described previously[47]. Briefly, mLN were mashed through a 40 µm cell strainer filter using a 5-ml syringe plunger, with the filtered cells representing the soluble cellular part and the remaining white matter left on strainer representing the stromal fraction. RNA was extracted with a Direct-zol RNA MiniPrep kit (Zymo Research) and reverse transcribed using RevertAid cDNA synthesis reagents (Thermo Scientific) for qPCR analysis. qPCR was performed using SYBR Green I Master Mix (Eurogentec) on an Applied Biosystems 7900HT System. Following primers were used to detect lymphangiogenic factors. *Vegf-a* forward: 5′- GCT GTA CCT CCA CCA TGC CAA G -3′, *Vegf-a* reverse: 5′- CGC ACT CCA GGG CTT CAT CG -3′, *Vegf-c* forward: 5′- GTG AGG TGT GTA TAG ATG TGG GG -3′, *Vegf-c* reverse: 5′- GTC TTG CTG AGG TAA CCT GTG -3′.

**Statistical analysis.** Statistical analyses were performed using a non-parametric Mann–Whitney Student's $t$ test, one-way or two-way ANOVA as indicated and with post-tests as appropriate. $P$-values indicated as $P < 0.05$ (*), $P < 0.01$ (**), $P < 0.001$ (***), $P < 0.0001$ (****) or ns (statistically not significant). Graph generation and statistical analyses were performed using Prism version 6 software (Graph pad, La Jolla, CA).

**Data availability.** All relevant data supporting the findings of this study are available within the paper or in Supplementary Files.

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

## Acknowledgements

We thank the École Polytechnique Fédérale de Lausanne (EPFL) animal facility, Miguel Garcia and the EPFL flow cytometry core facility, Jessica Sordet-Dessimoz and the histology core facility for support and expertise. A special thank you to Olivier Burri and Arne Seitz from the Bio-Imaging and Optics Platform for ImageJ/Fiji tools and crucial advice regarding image analysis. We also thank Biogen Idec for providing LTβR-Fc (to S.A.L.) for lymphotoxin studies. Finally, we would like to thank Leonardo Scarpellino and Chen-Ying-Yang from the University of Lausanne for valuable help in the generation of chimeric mice and Elke Scandella from the Kantonsspital St. Gallen for advice and help in genotyping of CCL19⁻ᶜʳᵉ × LTβRᶠˡ/ᶠˡ mice. This work was supported by the Leenaards prize for translational research in medicine awarded to N.L.H. and S.A.L. by the Leenaards Foundation, Lausanne, Switzerland.

## Author contributions

L.K.D. and N.L.H. conceived of and designed the study. L.K.D. performed all the experiments, analyzed the data, and wrote the paper. P.K. performed staining of some cryosections. B.L. provided *Ccl19⁻ᶜʳᵉ × LTβRᶠˡ/ᶠˡ* mice and critical suggestions at all steps of the manuscript. S.A.L. contributed *LT-β⁻/⁻* mice and critical suggestions throughout the manuscript. L.K.D. and N.L.H. analyzed the data and wrote the paper.

## Additional information

**Competing interests:** The authors declare no competing financial interests.

