## [Peer Review File · Nature Communications]

Reviewers' comments:

Reviewer #1 (Remarks to the Author):

This study by Lalit Kumar Dubey from the Harris lab is a great extension of previous work (Cell reports, May 2016) by the same authors and adds to the very productive and high quality work coming out of the Harris laboratory.

This study nicely identifies a central role played by LTB-producing, IL-4 and BAFFR-responsive, VEGF-producing B cells in communication with FRC's for reorganization of the mesenteric lymph nodes following GI worm infection.

The manuscript is well written, clear and laid out in a very logical and easy-to-follow manner. However, there are a few confusing areas, which are greatly helped by the schematic diagram at the end. What is hard to untangle from this study are the following:

1. Are the LTB-producing B cells, the same B cells that are responding to IL-4 and the same B cells that are making VEGF. i.e Are the B cells heterogeneous? Data or more discussion around this would be very interesting.
2. From the image shown in 3a, it is difficult to see how the graph in 3b was generated- particularly for naïve mice. It looks like there is a significant loss of Lyve-1 staining in LTB^{brfl} mice. This may be intensity (i.e. more in WT) giving the illusion of more. Can this be corrected?
3. I am not sure what value Figure 3c (showing CD11c+ cells) brings to this particular study. This interesting data is a little out of place and could be the foundation of future work - similar to data presented here as an extension from their previous Cell Reports work.
4. Is there a reason why WT mice were used as recipients in Fig4, rather than B cell deficient or T cell deficient recipient mice, which would be much cleaner. Is there an issue with lymph node architecture in using these mice as recipients? If so, this could be stated.
5. Data presented in Supl. Fig 4 comparing hematopoietic vs non- hematopoietic is interesting and needs more technical description.
 - a. How were compartments separated?
 - b. Are the authors sure that FRCs were not lost in processing?
 - c. What were the positive controls used for FRCs in the non- hematopoietic compartment?
 - d. Would IHC of tissue be more useful?
6. The FACS data in Supl. Fig 4b/c of VEGF producing cells is not very convincing. If the isotype is to be followed, then the gating strategy is inaccurate and most of the B cells in the naïve mLN would stain positive. If the authors want to make this claim- they either need to gate appropriately or add additional data to support this claim. At present it is an unnecessary blot on an otherwise very nice study.

Reviewer #2 (Remarks to the Author):

The manuscript by Dubey et al evaluates lymphangiogenesis in mesenteric lymph nodes of mice infected with a helminth, *Heligmosomoides polygyrus bakeri*. Previous publications by other authors

including Angeli et al and Liao and Ruddle (not noted here) have demonstrated a role for B cells in lymphangiogenesis after immunization and Angeli et al reported that B cells made VEGFA. That group (Tan et al) later reported that fibroblast reticular cells (FRCs) also produce VEGFC, a claim implicitly contradicted by the report by Dubey et al. The work reported here goes beyond the previous studies in that it implicates LT β R expressed by CCL19+ cells in this process and defines the interaction between B cells and FRCs. The authors conclude that LT α β produced by B cells stimulates FRC to make BAFF that in turns stimulates B cells that produce VEGFs that induce lymphangiogenesis. Many of these conclusions are supported by data but some are on shaky ground.

1. The authors' conclusions regarding the nature of the VEGF species are confusing. Most of the data rely on an ELISA that, according to the manufacturer, detects VEGFA. Figure 6 is very similar to the data of Angeli et al showing that B cells can make VEGFA in vitro after stimulation. The authors present data only in supplementary form that attempts to clarify this issue. In Supp Figure 1, they present RNA data for VEGFA and C and in supplementary figure 4, they stain B cells by FACS for VEGFC and VEGFA. Thus, they conclude that both VEGFs are important and that the source of VEGFC and A is B cells only. The authors need to be more precise in their discussion of the various VEGFs and acknowledge the contradiction to the Tan et al paper.

2. p.7 and Supplementary figure 2a-c. The authors state that Cre activity driven by CCL19 was restricted to FRCs. This is the conventional wisdom. However, in Figure sup 2c, there is YFP expression in cells that look like blood vessels. There have been reports that HEVs express CCL19 that is produced by FRCs and then transported across the vessel. What do the authors think is going on here? No matter what, I agree that CCL19 is not expressed by LYVE-1+ cells, but it does not look like CCL19 is "restricted" to FRCs.

3. Figure 3a is somewhat misleading. The graph shows no difference in area occupied by LECs in naïve wt and LTbRfl/fl but the figure shows a drastic difference. Which is it? This is an important point. The graph would suggest that the effect of LTbR is on inflammatory lymphangiogenesis. The photograph suggests it is on both homeostatic and inflammatory lymphangiogenesis.

4. Several figures utilize podoplanin to distinguish between lymphatic vessels and FRCs. In fact, both are pdn+. This is acknowledged occasionally in the manuscript, but not clearly stated and one is left with some uncertainty about whether these cells are being clearly distinguished.

Minor comments

1. p. 2 line 25 lead should be led
2. p.5 line 80. Insert The before Helminth
3. p.5 line 83 should be "tissue has not been studied in detail
4. p.5 line 87 should be: "infected mice and then.. "
5. p.5 one 93 should be "which were"
6. p.12 clarify that the mice are CCL19crexLTbRfl/fl in reference 19
7. p.12 line 241 Insert a reference for Liao and Ruddle, 2006 that predated ref. 35,36
8. p.21 should be LT β Rfl/f/
9. p.25 the complete reference for 18 is missing

Reviewer #3 (Remarks to the Author):

The study by Dubey et al., indicates that the crosstalk between FRC and B cells drives lymphangiogenesis in the expanded lymph node during helminth infection through a lymphotoxin receptor dependent- mechanism. The authors reveal the novel contribution of BAFF in stimulating VEGF-A production by B cells. Although the imaging in this study is well executed and beautiful, the novelty of the work is rather low and this is therefore a major weakness of the manuscript. Indeed, the novelty of this study is limited to the findings that helminth infection can induce lymph node lymphangiogenesis and the role of BAFF in this process since the implication of FRC, B cells and

lymphotoxin signaling in lymph node lymphangiogenesis has already been reported by several groups. Moreover, the following points need to be addressed by the authors.

1) Lymphangiogenesis was only measured by immunohistochemistry. The method used for quantification has not been provided. Furthermore, analysis of lymphangiogenesis should be further confirmed using flow cytometry which is more quantitative and analysis of lymphatic endothelial cell proliferation. The authors did not mention whether this lymphangiogenesis was accompanied by an expansion of FRCs?

2) Increased VEGF-C and VEGF-A transcripts were detected in infected LNs but what about the protein levels of these factors?

3) Fig 3A: LYVE-1 staining in naïve LN from $LT\beta R^{fl/fl}$ mice seems dimmer compared to $LT\beta R^{+/+}$ however, the quantification does not show any quantitative difference. This discrepancy should be addressed.

4) The authors should assess the expression of VEGF-A and VEGF-C in $LT\beta R^{fl/fl}$ mice and chimeric mice lacking $LT\beta$ in B cells. Do these two strains of mice exhibit changes in B cell proportion or number?

5) The authors indicate that BAFF is produced by stromal cells in particular FRCs in a $LT\beta R$ dependent manner which in a feedback loop stimulates B cells to produce VEGFs. The role of BAFF in promoting VEGFs in B cells is novel and therefore this finding should be further extended and confirmed. The authors should provide evidence that BAFF is necessary *in vivo* for VEGFs expression and lymphangiogenesis and should assess how mechanistically BAFF may support VEGFs production by B cells. Fig 6 shows the production of VEGF-A by B cells stimulated with BAFF \pm IL-4 but the data for VEGF-C is missing.

6) To follow the sequence of events described in Figure 7, the authors should number each event. BAFFR should be mentioned on B cells.

7) The authors omitted to cite the study by Shrestha B et al published in 2010 showing the role of VEGF-A derived B cells in driving lymphangiogenesis.

8) Fig S4: the legend has to be edited as panel A shows mRNA expression and not the flow cytometry gating strategy. The expression of VEGF-A and VEGF-C in B cells is not convincing in infected lymph node and the corresponding method is not described. Are B cells restimulated *in vitro* to induce the production of VEGF-A and VEGF-C? The non B cells fraction should be identified based on the absence of CD45 expression and expression of FRCs marker such as podoplanin.

Response to referees: Nature Communications 2017 (NCOMMS-17-00425)

Reviewer #1 (Remarks to the Author):

This study by Lalit Kumar Dubey from the Harris lab is a great extension of previous work (Cell reports, May 2016) by the same authors and adds to the very productive and high quality work coming out of the Harris laboratory. This study nicely identifies a central role played by LTB-producing, IL-4 and BAFFR-responsive, VEGF-producing B cells in communication with FRC's for reorganization of the mesenteric lymph nodes following GI worm infection. The manuscripts is well written, clear and laid out in a very logical and easy-to-follow manner. However, there are a few confusing areas, which are greatly helped by the schematic diagram at the end.

Response: We would like to thank the reviewer for their positive and encouraging comments about our work.

What is hard to untangle from this study are the following:

1. Are the LTB-producing B cells, the same B cells that are responding to IL-4 and the same B cells that are making VEGF. i.e Are the B cells heterogeneous? Data or more discussion around this are would be very interesting.

Response: We agree this is an interesting question and appreciate the suggestion. Unfortunately we weren't able to combine the VEGF and LTBR-Fc stains into the same panel, however we instead performed separate stains and included in a more extensive panel of B cell makers. Interestingly, we found that both VEGF and lymphotoxin expression were enriched within antigen experienced (IgD-) compared to naïve (IgD+) B cells indicating that their expression is associated with antigen

activation. This data has been included into the new Supplemental Figures 6 and described on page 9 lines 192-197. We think this information greatly enhances the manuscript and we have also included the need for BCR stimulation in the abstract (line 29), discussion (line 276) and the summary figure (now Figure 8).

2. From the image shown in 3a, it is difficult to see how the graph in 3b was generated- particularly for naïve mice. It looks like there is a significant loss of Lyve-1 staining in LTBrfl mice. This may be intensity (i.e. more in WT) giving the illusion of more. Can this be corrected?

Response: We would like to thank the reviewer for bringing this unintentional error to our attention. We re-evaluated the figure and found that indeed it was an intensity issue as pointed out by reviewer. We have now re-analyzed the data contained within the figure and included the new analysis in the revised manuscript. This, now correct, analysis shows that expression of Lyve-1 is comparable in WT and floxed naïve animals. For the reviewers interest we have also prepared a larger magnification image of the lymphatics in naïve mice (shown bellow; Response to Reviewer Figure-1).

Response to Reviewer Figure 1: CCL19^{cre} x LTβR^{fl/fl} (LTβR^{fl/fl}) and wildtype littermate control CCL19^{cre} x LTβR^{+/+} (LTβR^{+/+}) mice mLN serial cryosections showing lymphatics (Lyve-1: Red, DAPI: Blue) organization. Scale bar = 2000 μm. The insets were amplified and shown without DAPI to compare the signal intensity. Both mice showed a comparable Lyve-1 expression under homeostatic condition.

3. I am not sure what value Figure 3c (showing CD11c+ cells) brings to this particular study. This interesting data is a little out of place and could be the foundation of future work- similar to data presented here as an extension from their previous Cell Reports work.

Response: We agree with the reviewer that main focus of the manuscript is to delineate the mechanism of inflammatory lymphangiogenesis. However, the rationale behind including the CD11c+ cells was to strengthen our observation that lymphotoxin activation of FRCs is required to promote lymphangiogenesis, based on

the well-established need for lymphangiogenesis in promoting DC entry into the inflamed LN (Angeli et al., 2006). We also believe this finding strengthens the physiological relevance of our work and would therefore prefer to leave it in the manuscript.

4. Is there a reason why WT mice were used as recipients in Fig4, rather than B cell deficient or T cell deficient recipient mice, which would be much cleaner? Is there an issue with lymph node architecture in using these mice as recipients? If so, this could be stated.

Response: As speculated by the reviewer the rationale behind using WT mice as recipients is that these mice will have a normal lymphoid architecture along with fully developed, functional stromal and hematopoietic compartment prior to the irradiation. Previous studies also suggest that B cell-deficient mice exhibit altered secondary lymphoid organization (Nolte et al., 2004), along with several abnormalities involving macrophage and DC organization (Crowley et al., 1999) which is likely associated with alterations in the stromal cell populations that might remain after irradiation. Furthermore, T cell deficient animals lymph nodes have a higher density of Lyve-1⁺ lymphatic endothelial cells (Kataru et al., 2011). As per the reviewer's suggestion we have added a comment explaining the choice of WT mice as recipients to page 8, line 170-172 of the revised manuscript.

5. Data presented in Supl. Fig 4 comparing hematopoietic vs. non-hematopoietic is interesting and needs more technical description.

Response: We have updated the relevant information in materials and method section. Page 22 lines 496-500 in the manuscript

a. How were compartments separated?

Response: The hematopoietic and non- hematopoietic were separated using the widely accepted protocol first published in PNAS 2014, 111 (1) E109-E118. In brief the lymph nodes were smashed through a 40µm cell strainer and the soluble fraction collected as the hematopoietic cells, whilst the remaining white matter left on cell the strainer was collected as the non-hematopoietic fraction (stromal cells). We have updated the text and added the indicated reference to the materials and methods section page 22 lines 496-500.

b. Are the authors sure that FRCs were not lost in processing?

Response: To confirm that FRCs were not lost in the processing we performed flow cytometry on both the cellular fraction and the stromal fraction (collected as described above). We confirmed the published findings, observing that CD45-pdnp+CD31-cells (FRCs) were present in the white matter collected from the cell strainer (stromal fraction), but were not found in the hematopoietic fraction. By contrast we could readily detect CD45+ B and T cells in the hematopoietic fraction but not in the stromal fraction. These findings were also confirmed by the gene expression analysis outlined in our answer to question (c) below.

c. What were the positive controls used for FRCs in the non- hematopoietic compartment?

Response: In response to the reviewers question we performed RT-PCR for genes that are known to be associated only with stromal cells (CCL19 and IL-7) or hematopoietic cells (IL-4 and IL-13). The results are shown below (Response to Reviewer Figure-2) for the reviewer’s interest and confirm that our method separates the stromal and hematopoietic cells.

Response to Reviewer Figure 2: Relative expression of mRNA encoding CCL-19, IL-7, IL-4 and IL-13 in mLN stromal vs. cellular fraction were determined over the indicated time point. Relative expression was calculated using $2^{-\Delta\Delta Ct}$ methods. Samples with Ct value above 33 were not considered for analysis. CCL19 and IL-7 expression were detectable in stromal fraction where as IL-4 and IL-13 expression was seen in only cellular fraction.

d. Would IHC of tissue be more useful?

Response: We agree with the reviewer and we invested a substantial amount of time and effort into establishing the IHC detection of VEGFs during the course of this

project. In our initial attempts we found convincing expression of VEGF-A by IHC, and this was associated with B cell follicles (see the image Response to Reviewer Figure-3). Unfortunately however we found that the staining's were not reproducible and, although we tested a number of different anti-VEGF antibodies, we repeatedly encountered serious problems with the background. For this reason we have elected not to show IHC and to instead rely solely on detection of mRNA transcripts by realtime-PCR or protein by ELISA and flow cytometry, all of which generated reproducible results in our hands.

Response to Reviewer Figure 3: mLN VEGF-A expression during *Hpb* infected mice. Green: B220 staining, Red: VEGF-A staining. Images were acquired using Olympus slide scanner. The Overlay panel indicates that VEGF-A expression was largely restricted to B cell compartment.

6. The FACS data in Supl. Fig 4b/c of VEGF producing cells is not very convincing. If the isotype is to be followed, then the gating strategy is inaccurate and most of the B cells in the naïve mLN would stain positive. If the authors want to make this claim- they either need to gate appropriately or add additional data to support this claim. At present it is an unnecessary blot on an otherwise very nice study.

Response: We would like to thank the reviewer for pointing out the problems in the gating strategy used. We have now performed further repeats of this experiment to validate our results and to perform a more in depth analysis of the B cell populations making VEGF (in response to question 1). The new data has been incorporated into the manuscript as the new Supplemental Figures 6 and described on page 9 lines 190-198. We believe that these new data, incorporating more careful gating strategies, clearly support our conclusion that B cells, and in particular activated B cells, are the major source of VEGFA&C in the mLN of helminth infected mice.

Reviewer #2 (Remarks to the Author):

The manuscript by Dubey et al evaluates lymphangiogenesis in mesenteric lymph nodes of mice infected with a helminth, Heligmosomoides polygyrus bakeri. Previous publications by other authors including Angeli et al and Liao and Ruddle (not noted here) have demonstrated a role for B cells in lymphangiogenesis after immunization and Angeli et al reported that B cells made VEGFA. That group (Tan et al) later reported that fibroblast reticular cells (FRCs) also produce VEGFC, a claim implicitly contradicted by the report by Dubey et al. The work reported here goes beyond the previous studies in that it implicates LTβR expressed by CCL19+ cells in this process and defines the interaction between B cells and FRCs. The authors conclude that LTαβ produced by B cells stimulates FRC to make BAFF that in turns stimulates B cells that produce VEGFs that induce lymphangiogenesis. Many of these conclusions are supported by data but some are

on shaky ground.

Response: We would like to thank the reviewer for their positive comments about our findings and for pointing out where he/she felt there were weaknesses in the conclusions. We have now provided extensive new experimental data to address these concerns and feel that this has greatly improved the manuscript.

The authors' conclusions regarding the nature of the VEGF species are confusing. Most of the data rely on an ELISA that, according to the manufacturer, detects VEGFA. Figure 6 is very similar to the data of Angeli et al showing that B cells can make VEGFA in vitro after stimulation. The authors present data only in supplementary form that attempts to clarify this issue. In Supp Figure 1, they present RNA data for VEGFA and C and in supplementary figure 4, they stain B cells by FACS for VEGFC and VEGFA. Thus, they conclude that both VEGFs are important and that the source of VEGFC and A is B cells only. The authors need to be more precise in their discussion of the various VEGFs and acknowledge the contradiction to the Tan et al paper.

Response: We would like to thank the reviewer for pointing out the confusion resulting from our discussion of the VEGF factors. We have now addressed this concern by performing separate analyses of VEGF-A and VEGF-C in all of our assays (RT-PCR; FACs and ELISA). The new ELISA data for VEGF-C has been added to Suppl Fig 1 (ELISA On tissue homogenates) and the main Figure 6 (ELISA on B cell culture supernatants). We have also made revisions throughout the entire manuscript to call out VEGF-A and VEGF-C separately and additionally referred to

published work detailing a role for VEGF-A or VEGF-C in lymphatic vessel growth to the introduction (references 2,12,13,14, page 3, lines 46-50). Lastly we have addressed our data in the context of the manuscript by Tan et.al. paper (see page 13-14, lines 283-294). We do not feel our data contradict this manuscript, indeed we clearly see that stromal cells are capable of making VEGF-A & C. Rather our work shows that in response to helminth infection, B cells and not stromal cells represent the major source of increased VEGFA/C production. We have made a more careful effort in the revised manuscript to indicate that whilst this is true for helminth-induced lymphangiogenesis, other inflammatory triggers seem to elicit increased VEGF production by stromal cells (see page 13-14, line 283-294 of the discussion for our comments in this regard).

2. p.7 and Supplementary figure 2a-c. The authors state that Cre activity driven by CCL19 was restricted to FRCs. This is the conventional wisdom. However, in Figure sup 2c, there is YFP expression in cells that look like blood vessels. There have been reports that HEVs express CCL19 that is produced by FRCs and then transported across the vessel. What do the authors think is going on here? No matter what, I agree that CCL19 is not expressed by LYVE-1+ cells, but it does not look like CCL19 is “restricted” to FRCs.

Response: We agree with the reviewer that the YPF expression in cells contained within Figure 2c have the morphology of blood vessels and in response to their comments we have performed a more extensive analysis of the cells in this area and included the use of markers specific for endothelial cells (CD31). From this work we observed that CD31+ cells within blood endothelial cells and within high endothelial

venules (HEVs) were not YFP positive, however (and of great interest) YFP+ cells were aligned very closely to the endothelial cells. These cells are most likely FRCs or a cell type closely related to FRCs. We have now included the images showing the CD31 staining's, plus higher magnification images of the endothelial cells, within supplementary figure 3 of the revised manuscript. A description of these findings has also been added to page 7-8 lines 154-156 of the main text.

3. Figure 3a is somewhat misleading. The graph shows no difference in area occupied by LECs in naïve wt and LTbRfl/fl but the figure shows a drastic difference. Which is it? This is an important point. The graph would suggest that the effect of LTbR is on inflammatory lymphangiogenesis. The photograph suggests it is on both homeostatic and inflammatory lymphangiogenesis.

Response: We would like to thank the reviewer for bringing this problem to our attention. We would like to thank the reviewer for bringing this unintentional error to our attention. We re-evaluated the figure and found that it was an intensity issue as pointed out by reviewer. We have now re-analyzed the data contained within the figure and included the new analysis in the revised manuscript. This, now correct, analysis shows that expression of Lyve-1 is comparable in WT and floxed naïve animals. For the reviewers interest we have also prepared a larger magnification image of the lymphatics in naïve mice (shown in the response to reviewer 1, question 2).

4. Several figures utilize podoplanin to distinguish between lymphatic vessels and FRCs. In fact, both are pdn+. This is acknowledged occasionally in the manuscript,

but not clearly stated and one is left with some uncertainty about whether these cells are being clearly distinguished.

Response: Based on the reviewers comments we have included a more full description of the markers we used to distinguish LECs and FRCs. We have highlighted the fact that both FRCs and LECs are pdpn+ on page 6 lines 119-122 of the manuscript and explained that the Lyve-1+ (which is specific for LECs) is used to differentiate the two cell types (page 6, lines 121-122). We have also explained that ERTR7 (used in Figure 2) is a FRC specific marker (page 7 line 129-131). We feel that these changes have greatly improved the clarity of the manuscript and would like to thank the reviewer for pointing out the ambiguity raised by the absence of these explanations.

Minor comments (Corrected in the manuscript)

1. p. 2 line 25 lead should be led

We have exchanged the word as suggested in the text.

2. p.5 line 80. Insert The before Helminth

We have inserted the word as suggested in the text.

3. p.5 line 83 should be “tissue has not been studied in detail

We have exchanged the word as suggested in the text

4. p.5 line 87 should be: ”infected mice and then.. “

This has now been corrected in the text.

5. p.5 one 93 should be “which were”

This has now been corrected in the text.

6. p.12 clarify that the mice are CCL19crexLTbRfl/fl in reference 19

This has now been corrected in the text.

7. p.12 line 241 Insert a reference for Liao and Ruddle, 2006 that predated ref. 35,36

The suggested reference is cited in the manuscript (Ref: 40).

8. p.21 should be $LT\beta Rfl/f$

This has now been corrected in the text.

9. p.25 the complete reference for 18 is missing

We thank reviewer for pointing out this mistake. We have corrected the reference.

Reviewer #3 (Remarks to the Author):

The study by Dubey et al., indicates that the crosstalk between FRC and B cells drives lymphangiogenesis in the expanded lymph node during helminth infection through a lymphotoxin receptor dependent- mechanism. The authors reveal the novel contribution of BAFF in stimulating VEGF-A production by B cells. Although the imaging in this study is well executed and beautiful, the novelty of the work is rather low and this is therefore a major weakness of the manuscript. Indeed, the novelty of this study is limited to the findings that helminth infection can induce lymph node lymphangiogenesis and the role of BAFF in this process since the implication of FRC, B cells and lymphotoxin signaling in lymph node lymphangiogenesis has already been reported by several groups. Moreover, the following points need to be addressed by the authors.

Response: We would like to thank the reviewer for their encouraging comments and

appreciation of the quality of the imaging. We agree that our finding that BAFF promotes lymphangiogenesis represents the most novel and exciting part of this work. Based on the reviewers suggestions we have expanded our work related to BAFF and provide a substantial amount of new data showing that in vivo blockade of BAFF attenuated VEGF-A and VEGF-C production and associated lymphangiogenesis following helminth infection (see response to question 5 below for full details).

1) Lymphangiogenesis was only measured by immunohistochemistry. The method used for quantification has not been provided.

Response: We have now included an explanation of how our quantitative analysis of lymphangiogenesis by immunohistochemistry (figure 1c and 3b) was performed (page 20, lines 442-447).

Furthermore, analysis of lymphangiogenesis should be further confirmed using flow cytometry, which is more quantitative, and analysis of lymphatic endothelial cell proliferation. The authors did not mention whether this lymphangiogenesis was accompanied by an expansion of FRCs?

Response: We agree with the reviewer that quantitation by flow cytometry is more accurate and has the added benefit of allowing an assessment of proliferation. Based on the reviewers comments we have performed new experiments in which we assessed the numbers of FRCs and LECs present in naïve and infected mice using the widely accepted protocol of pdpn and CD31 staining (Cremasco et al., 2014; Link et al., 2007). We also included Ki-67 staining to assess proliferation. These data show that both FRCs and LECs increased in total number following infection and that expansion is associated with an increase in the number and proportion of proliferating

cells. This data is now shown as the new supplementary figure-2 (a, b, c) and is described in the text of the main manuscript (page 7, lines 132-137). The materials and methods section has also been updated and includes the relevant reference (page 18, lines 381-383).

2) Increased VEGF-C and VEGF-A transcripts were detected in infected LNs but what about the protein levels of this factor?

Response: We would like to thank the reviewer for suggesting this experiment. Based on their comment we have now assessed VEGF-A and VEGF-C protein levels in mLN tissue homogenates prepared from naïve and helminth infected mice using ELISA. The results show clearly increased levels of both VEGF-A and VEGF-C in response to infection and are incorporated into the manuscript as Supplementary Figure 1h-i. A description of these data has also been added to the main text (page 5-6, line 102-106).

3) Fig 3A: LYVE-1 staining in naïve LN from $LT\alpha Rfl/fl$ mice seems dimmer compared to $LTBR^{+/+}$ however, the quantification does not show any quantitative difference. This discrepancy should be addressed.

Response: We would like to thank the reviewer for bringing this problem to our attention. We re-evaluated the figure and found that we had accidentally used two distinct shades of red from the LUT panel used in the Fiji program. We have now re-analyzed the data contained within the figure using the same LUT colors and included the new analysis in the revised manuscript. This, now correct, analysis shows that

expression of Lyve-1 is comparable in WT and floxed naïve animals. For reviewers interest we have also prepared a larger magnification image of the lymphatics in naïve mice (shown in the response to reviewer 1, question 2).

4) The authors should assess the expression of VEGF-A and VEGF-C in $LT\beta^{fl/fl}$ mice and chimeric mice lacking $LT\beta$ in B cells.

Response: We only have tissues collected for immunohistochemistry from these mice and our attempts to reproducibly and accurately assess VEGF-A and VEGF-C by IHC were not successful (please see response to reviewer 1, question 5d). We would therefore need to assess VEGF-A/C production in these mice by RT-PCR or ELISA. However, and very unfortunately, serious problems with current breeding of the $LT\beta^{fl/fl}$ mice preclude us from being able to complete this experiment at the current time. Moreover repeating the BMC experiment for the purpose of these measurements would take 4-5 months, which is well beyond the 3 month timeframe we were asked to complete our revisions within. We hope that the reviewer can appreciate these limitations. We also hope that they agree with us that our findings showing an absence of lymphangiogenesis in $LT\beta^{fl/fl}$ mice and BMCs in which B cells lack $LT\beta$ (Figures 3 and 4) is adequate evidence that VEGF production is altered, based on the previous publications showing that VEGF-A and C are necessary for lymphangiogenesis (Alitalo, 2011; Angeli et al., 2006; Tammela and Alitalo, 2010). Instead the focus of our work was to show that helminth infection upregulates both VEGF-A and C production (Supplementary Figure 1) and to identify the source of these cytokines as being B cells (Supplementary Figures 5 and 6).

Do these two strains of mice exhibit changes in B cell proportion or number?

Response: The numbers of B cells present in the $LT\beta R^{fl/fl}$ mice and chimeric mice (lacking $LT\alpha$ in B cells) was assessed by counting the number of B cell follicles present in IHC stains of whole mLNs. Both sets of mice exhibit normal numbers of B cell follicles in the naïve state, however they fail to generate new B cell follicles in response to helminth infection, in keeping with our previous report detailing a role for B cell-derived lymphotoxin signaling to FRCs in the formation of new follicles (Dubey et al., 2016). The data for the $LT\beta R^{fl/fl}$ mice has already been shown in our previous publication (Dubey et al., 2016) and the data for the chimeric mice is shown below for the reviewer’s interest (Response to Reviewer Figure-4). We have also added a statement about these findings to page 8, lines 174-181 of the text of our revised manuscript.

Response to Reviewer Figure 4: Histogram showing number of B cell follicles in chimeric mice mLN lacking lymphotoxin expression on B cells (B-LTb) or T cells (T-LTb). Follicles were counted at high magnification using a graphic tablet based on B220+ staining. For each group, data is represented as mean \pm SEM. Naïve mLN:

n=2, 21dpi HP mLN: n=3. B-WT and T-WT represents the control groups where B and T cells are lymphotoxin sufficient.

5) The authors indicate that BAFF is produced by stromal cells in particular FRCs in a LT α R dependent manner which in a feedback loop stimulates B cells to produce VEGFs. The role of BAFF in promoting VEGFs in B cells is novel and therefore this finding should be further extended and confirmed. The authors should provide evidence that BAFF is necessary in vivo for VEGFs expression and lymphangiogenesis and should assess how mechanistically BAFF may support VEGFs production by B cells.

Response: We agree with the reviewer that the role for BAFF in promoting VEGF production by B cells represents one of the most exciting advances in our study and that this deserves to be expanded. Based on their comments we have now performed new experiments in which we blocked the function of BAFF in vivo using a neutralizing mAb (Sandy-2). We were able to confirm previous studies showing that this antibody blocks BAFF function in vivo (Supplementary Figure 8) and additionally performed experiments using helminth-infected mice. Our data show that anti-BAFF mAb treatment attenuates the expansion of B cells, FRCs and LECs (as assessed by flow cytometry) following helminth infection and that this results in a strong decrease in lymphangiogenesis (as assessed by immunohistochemistry). We also observed that anti-BAFF mAb treatment strongly reduced VEGF-A production and abrogated VEGFC production in the mLN of infected mice. These data have been included in the manuscript as a new Figure 7 and a new Supplementary Figure 8. A full description of our findings has also been added to the text under the subheading ‘Administration of a neutralizing anti-BAFF mAb attenuates helminth-induced mLN

lymphangiogenesis' page 11, lines 235-253.

Fig 6 shows the production of VEGF-A by B cell stimulated with BAFF+/- IL-4 but the data for VEGF-C is missing.

Response: We agree with the reviewer that it would be useful to include an analysis of VEGF-C and we have now performed the necessary ELISAs. These data show that BAFF together with IL-4 also elicits the production of VEGF-C by anti-IgM stimulated B cells and we have added the new data to Figure 6 of the manuscript. A description of these findings has also been added to the text, page 10, lines 223-224.

We would like to thank the reviewer for their suggestions regarding the new experiments to be performed and we believe that the new data resulting from these experiments greatly strengthens the manuscript.

6) To follow the sequence of events described in Figure 7, the authors should number each event. BAFFR should be mentioned on B cells.

Response: We agree and we have now incorporated numbering into the summary figure (now Figure 8) and the corresponding legend.

7) The authors omitted to cite the study by Shrestha B et al published in 2010 showing the role of VEGF-A derived B cells in driving lymphangiogenesis.

Response: We would like to thank the reviewer for pointing out this oversight and we

have now included this reference (Ref No: 13) into the introduction (see page 3, line 50 and 55) and in discussion sections (page 15, line 328-329) of the revised manuscript.

8) Fig S4: the legend has to be edited as panel A shows mRNA expression and not the flow cytometry gating strategy.

Response: Our records show that the legend to Fig S4 panel A was correct and called out mRNA expression not flow cytometry, this has therefore not been altered.

The expression of VEGF-A and VEGF-C in B cells is not convincing in infected lymph node and the corresponding method is not described. Are B cells restimulated in vitro to induce the production of VEGF-A and VEGF-C? The non-B cells fraction should be identified based on the absence of CD45 expression and expression of FRCs marker such as podoplanin.

Response: In response to the reviewer's comments, and the comments of reviewer 1, we have repeated the experiments shown in the original Fig S4 to validate our results and to perform a more in depth analysis of the stromal and B cell populations making VEGF-A and -C. The new data has been incorporated into the manuscript as the new Supplemental Figures 6 and described on page 9 lines 190-198. We have also added a description of the methods used on page 21 lines 474-483 of the revised manuscript. We believe that these new data, incorporating more careful gating strategies, clearly support our conclusions that B cells, and in particular activated B cells, are the major source of VEGF-A and VEGF-C in the mLN of helminth infected mice (B cells were

not re-stimulated prior to these stainings). The data does show some degree of VEGF-A and VEGF-C production by the stromal cell fraction (identified as CD45-ve), however this does not increase following helminth infection. The finding (both by RT-PCR and Flow cytometry) that stromal cells can make VEGF-A/C has been called out in the revised manuscript and we have also added a discussion of publications that show that FRCs rather than B cells represent an important source of increased VEGF production in response to other forms of stimuli (Ref 29, 36, page 13-14, lines 283-294).

References:

- Alitalo, K. (2011). The lymphatic vasculature in disease. *Nat Med* *17*, 1371-1380.
- Angeli, V., Ginhoux, F., Llodrà, J., Quemeneur, L., Frenette, P.S., Skobe, M., Jessberger, R., Merad, M., and Randolph, G.J. (2006). B Cell-Driven Lymphangiogenesis in Inflamed Lymph Nodes Enhances Dendritic Cell Mobilization. *Immunity* *24*, 203-215.
- Cremasco, V., Woodruff, M.C., Onder, L., Cupovic, J., Nieves-Bonilla, J.M., Schildberg, F.A., Chang, J., Cremasco, F., Harvey, C.J., Wucherpfennig, K., *et al.* (2014). B cell homeostasis and follicle confines are governed by fibroblastic reticular cells. *Nat Immunol* *15*, 973-981.
- Crowley, M.T., Reilly, C.R., and Lo, D. (1999). Influence of Lymphocytes on the Presence and Organization of Dendritic Cell Subsets in the Spleen. *The Journal of Immunology* *163*, 4894-4900.

Kataru, R.P., Kim, H., Jang, C., Choi, D.K., Koh, B.I., Kim, M., Gollamudi, S., Kim, Y.-K., Lee, S.-H., and Koh, G.Y. (2011). T Lymphocytes Negatively Regulate Lymph Node Lymphatic Vessel Formation. *Immunity* 34, 96-107.

Link, A., Vogt, T.K., Favre, S., Britschgi, M.R., Acha-Orbea, H., Hinz, B., Cyster, J.G., and Luther, S.A. (2007). Fibroblastic reticular cells in lymph nodes regulate the homeostasis of naive T cells. *Nat Immunol* 8, 1255-1265.

Nolte, M.A., Arens, R., Kraus, M., van Oers, M.H.J., Kraal, G., van Lier, R.A.W., and Mebius, R.E. (2004). B Cells Are Crucial for Both Development and Maintenance of the Splenic Marginal Zone. *The Journal of Immunology* 172, 3620-3627.

Tammela, T., and Alitalo, K. (2010). Lymphangiogenesis: Molecular Mechanisms and Future Promise. *Cell* 140, 460-476.

REVIEWERS' COMMENTS:

Reviewer #1 (Remarks to the Author):

The authors have thoroughly and satisfactorily addressed all of my initial concerns. This is a very nice study and will be of interest to a wide readership.

Reviewer #2 (Remarks to the Author):

The manuscript by Dubey et al evaluates lymphangiogenesis in mesenteric lymph nodes of mice infected with a helminth, *Heligmosomoides polygyrus bakeri*. The work reported here goes beyond the previous studies in that it implicates LT β R expressed by CCL19+ cells in this process and defines the interaction between B cells and FRCs. The authors conclude that LT $\alpha\beta$ produced by B cells stimulates FRC to make BAFF that in turns stimulates B cells that produce VEGFs that induce lymphangiogenesis. These conclusions are well justified by the extensive studies involving, staining, PCR and ELISAs. The authors do a nice job of distinguishing between VEGF-A and VEGF-C. The discussion and data support the concept that the context is particularly relevant when evaluating regulation of lymphangiogenesis.

Reviewer #3 (Remarks to the Author):

The authors have addressed all the comments. No further concerns.

Response to referees 2: Nature Communications 2017 (NCOMMS-17-00425)

Reviewer #1 (Remarks to the Author):

The authors have thoroughly and satisfactorily addressed all of my initial concerns. This is a very nice study and will be of interest to a wide readership.

Reviewer #2 (Remarks to the Author):

The manuscript by Dubey et al evaluates lymphangiogenesis in mesenteric lymph nodes of mice infected with a helminth, *Heligmosomoides polygyrus bakeri*. The work reported here goes beyond the previous studies in that it implicates $LT\beta R$ expressed by CCL19+ cells in this process and defines the interaction between B cells and FRCs. The authors conclude that $LT\alpha\beta$ produced by B cells stimulates FRC to make BAFF that in turns stimulates B cells that produce VEGFs that induce lymphangiogenesis. These conclusions are well justified by the extensive studies involving, staining, PCR and ELISAs. The authors do a nice job of distinguishing between VEGF-A and VEGF-C. The discussion and data support the concept that the context is particularly relevant when evaluating regulation of lymphangiogenesis.

Reviewer #3 (Remarks to the Author):

The authors have addressed all the comments. No further concerns.

Author's response:

We thank all the three reviewers for their positive comments and endorsing our manuscript for publication. Thank you.